# Is On-Policy Data always the Best Choice for Direct Preference Optimization-based LM Alignment?

**Zetian Sun, Dongfang Li, Xuhui Chen, Baotian Hu,**[*] **Min Zhang**
Harbin Institute of Technology, Shenzhen
zetiansun.cs@gmail.com
{lidongfang, hubaotian, zhangmin2021}@hit.edu.cn

## Abstract

The alignment of language models (LMs) with human preferences is critical for building reliable AI systems. The problem is typically framed as optimizing an LM policy to maximize the expected reward that reflects human preferences. Recently, Direct Preference Optimization (DPO) was proposed as an LM alignment method that directly optimizes the policy from static preference data, and further improved by incorporating on-policy sampling (i.e., preference candidates generated during the training loop) for better LM alignment. However, we show on-policy data is not always optimal, with systematic effectiveness difference emerging between static and on-policy preference candidates. For example, on-policy data can result in a $3\times$ effectiveness compared with static data for Llama-3, and a $0.4\times$ effectiveness for Zephyr. To explain the phenomenon, we propose the alignment stage assumption, which divides the alignment process into two distinct stages: the preference injection stage, which benefits from diverse data, and the preference fine-tuning stage, which favors high-quality data. Through theoretical and empirical analysis, we characterize these stages and propose an effective algorithm to identify the boundaries between them. We perform experiments on 5 models (Llama, Zephyr, Phi-2, Qwen, Pythia) and 2 alignment methods (DPO, SLiC-HF) to show the generalizability of alignment stage assumption and the effectiveness of the boundary measurement algorithm.

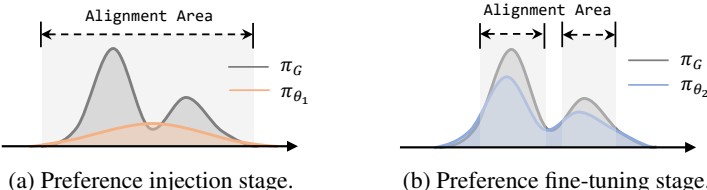

(a) Preference injection stage.  (b) Preference fine-tuning stage.

Figure 1: Illustration of our alignment stage assumption and different characteristics of (a) preference injection stage and (b) preference fine-tuning stage. The alignment area indicates the preferred region of preference candidates at corresponding alignment stages. The stage boundary is estimated by the distance between ground truth text distribution ($\pi_G$) and simulated text distribution ($\pi_{\theta_1}, \pi_{\theta_2}$).

## 1 Introduction

Large language models possess broad world knowledge and strong generalization capabilities in complex tasks under minimal supervision (Brown et al., 2020). However, the powerful models still produce biased (Bender et al., 2021), unfaithful (Ji et al., 2023) and harmful (Bai et al., 2022) responses due to the heterogeneous sources of their pre-training corpora. It is important to ensure models to generate desired responses that conform to humans' ethical standards and quality preferences for building reliable AI systems, which is well known as language model (LM) alignment

---

[*]Corresponding Author

with human preferences (Ouyang et al., 2022). Generally, the LM alignment problem is formulated as optimizing a policy model $\pi_\theta$ to maximize the expected reward $r_\phi$, where the reward $r_\phi$ reflects human preference regarding the completion $y$ for a given prompt $x$.

The most widely adopted approach to address the LM alignment problem is through reinforcement learning (RL) in an **on-policy** manner (Ziegler et al., 2019; Stiennon et al., 2020; Ouyang et al., 2022). Specifically, the on-policy manner requires $\pi_\theta$ iteratively refines its policy by performing on-policy sampling (i.e., sampling completions generated under its current parameters), ensuring that gradient estimates align with the latest behavior policy. The LM policy is then optimized via RL solutions. However, these approaches incur significant computational cost due to repeated sampling from the LM policy, and are observed to be unstable due to the high variance in estimating the policy gradients or value functions, which potentially worsens sample complexity and thus compromises efficient model convergence (Papini et al., 2018; Anschel et al., 2017).

Direct Preference Optimization (DPO, Rafailov et al. (2023)) was proposed to be a competitive alternative to the RL solutions. Specifically, DPO optimizes $\pi_\theta$ via reward modeling loss on preference candidates following the **off-policy** manner, i.e., the LM policy is optimized on a static dataset without additional sampling during the training loop. It is more resource-efficient, and shares the theoretically equivalent optimization objective with those RL solutions. Despite all the advantages, as an off-policy method, DPO can struggle in out-of-distribution scenarios and result in sub-optimal performance due to the absence of on-policy exploration (Tang et al., 2024).

To tackle these issues, recent works proposed iterative DPO, a method integrating on-policy sampling into regular DPO training, as observed to outperform vanilla DPO in several benchmarks (Wu et al., 2024; Zhang et al., 2025a; Rosset et al., 2024). These findings highlight the potential of on-policy sampling for enhancing LM alignment via off-policy methods like DPO. However, the practical recipe of using on-policy data lacks discussion or clear guidelines. Several works choose to train the LM policy on on-policy data directly (Yuan et al., 2024; Liu et al., 2024), while other works choose to train models on off-policy preference candidates first as a cold start phase (Zhang et al., 2025a; Kim et al., 2025). Such discrepancy and arbitrariness indicate an absence of comprehensive understanding about the relationship between LM alignment and preference candidates, which may limit the model performance and sample efficiency. This motivates us to study the following research question: ***What is the requirement of preference candidates during the LM alignment process?*** In this work, we answer the research question from two aspects, i.e, the qualitative description of the LM alignment process (`RQ1`) and the actionable insight of the description of the LM alignment process (`RQ2`). Through detailed experiments, we reveal a patterned dynamic requirements of preference candidates during the alignment process, and further provide an alignment stage assumption to explain the phenomenon from the perspective of DPO. Based on the assumption, we answer RQs through massive empirical experiments and theoretical-grounded method.

Firstly, we conduct a two-iteration training experiment on Llama-3, Zephyr and Phi-2. The experimental results reveal the existence of a patterned effectiveness discrepancy between the use of on-policy preference candidates ($PC_{on}$) and off-policy preference candidates ($PC_{off}$), and models exhibit varying performances and dynamic requirements for preference candidates. Motivated by this observation, we propose the *alignment stage assumption*, which posits that the alignment process can be divided into two stages, i.e., the preference injection stage and the preference fine-tuning stage, as illustrated in Figure 1. Based on the alignment stage assumption, we answer the research questions subsequently. Specifically, we conduct extensive experiments to demonstrate the characteristics of each alignment stage (for `RQ1`). We find that models in preference injection stage favor data of high preference diversity, while those in preference fine-tuning stage favor data of high preference quality. We propose the boundary measurement algorithm, a measurement to determine which stage the policy is currently in, and perform extensive experiments to show the effectiveness of our algorithm (for `RQ2`). Moreover, we provide a theoretical perspective to interpret the stage characteristics and the boundary measurement algorithm. Notably, we show that the requirements of preference diversity stems from a more accurate approximation of the ground-truth preference given the Bradley-Terry definition. The goal of selecting preference candidates is to better estimate the general text distribution, which is based on human preferences or the ground-truth reward model used for preference annotation. We also show that our boundary measurement algorithm identifies a better estimation of the general text distribution. Finally, we conduct experiments on more models (Qwen 2.5, Pythia) and more methods (SLiC-HF) to show the generalizability of our conclusions. To provide a clear image, we illustrate the assumption and its subsequent conclusions in Figure 2.

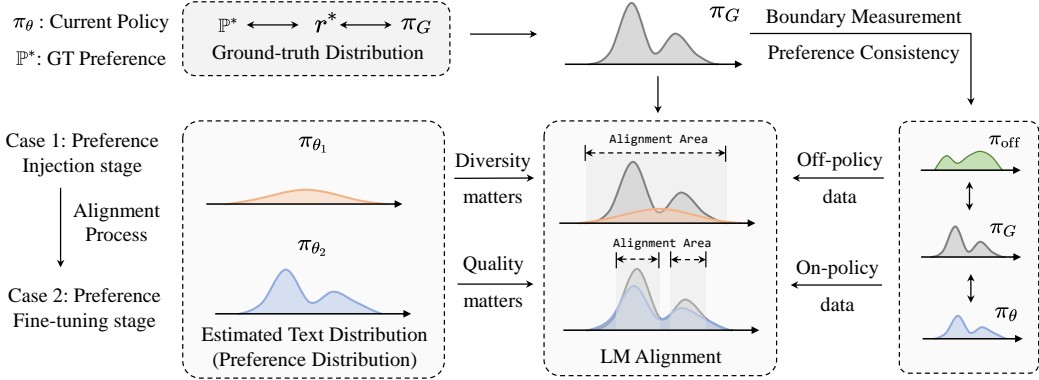

Figure 2: Illustration of the alignment stage assumption. The alignment process is a continuous transition from preference injection stage to preference fine-tuning stage. We demonstrate the characteristics of stages (Case 1 and Case 2). We build up the relationship among preference distribution, reward model and text distribution, which help us understand the alignment process from the perspective of distribution distance and preference consistency. Practically, we propose the boundary measurement, a measurement to decide which stage the policy is currently in by judging which distribution ($\pi_{\mathrm{off}}$ and $\pi_\theta$) is a better estimation of the ground-truth distribution ($\pi_G$).

We summarize our contributions in this paper:

- We reveal a patterned effectiveness discrepancy between on-policy data and off-policy for different models, and propose the alignment stage assumption (preference injection stage, preference fine-tuning stage) to model the dynamic requirements for preference candidates.
- We analyze the alignment stages through empirical analysis on two characteristics (i.e., diversity and quality), showing that models in preference injection stage favor data with high diversity, while models in preference fine-tuning stage favor data with high quality.
- We provide theoretical insights into the underlying mechanism about the LM alignment process, and propose the boundary measurement algorithm to decide stage boundaries.

## 2 RELATED WORK

**Iterative DPO.** Based on vanilla DPO, iterative DPO aims at improving DPO by incorporating on-policy sampling data. Yuan et al. (2024) constructs the preference dataset automatically where both preference candidates and instruction prompts are generated by LM in an on-policy manner. Tajwar et al. (2024) further discusses the requirements of fine-tuning with preference data through extensive experiments and detailed theoretical analysis, showing that approaches that use on-policy sampling are generally more preferred in practice. These works provide theoretical analysis about on-policy sampling. Our work builds on this line by describing the overall alignment process from a systematic and methodological perspective and improving the efficiency and effectiveness of on-policy sampling for model training, rather than selecting preference data manually and empirically, which is neither scalable nor optimal for LM alignment.

**Data Diversity.** The diversity of preference data can be separated into two sections: preference diversity and candidate diversity, both facts can help improve LM alignment. The former is due to the complexity of values, environments or populations, which result in the mismatch and diversity of preferences among different annotators. Several works model the diverse preference alignment problem as a multi-object optimization problem, addressing the problem using methods like Pareto optimality (Guo et al., 2024; Zhou et al., 2024) or reward ensembling (Lou et al., 2024; Zeng et al., 2024; Ramé et al., 2024). Our work focuses on the latter one, the candidate diversity. It is due to the limited coverage of the general text space given the condition of finite sampling, which results in an insufficient and incomplete preference representation. By labeling preferences using the same reward model, our work introduces the crucial role of candidate diversity at the preference injection stage. It can help models construct the general reward distribution effectively that is aligned with the reward model, and thus achieve more valuable explorations at the preference fine-tuning stage.

## 3 PRELIMINARIES

In this section, we first formally review the concept and objective of the language model alignment problem. Then we review existing approaches that are applied to address the alignment problem via reinforcement learning and direct preference optimization.

### 3.1 LM ALIGNMENT WITH HUMAN PREFERENCES

Given a vocabulary $\mathcal{V}$, a language model defines a probability distribution $\pi(x) = \prod_{t=1}^{n} \pi(x_t|x_1, ..., x_{t-1})$ over a sequence of tokens $x = (x_1, ..., x_n)$. We apply $\pi$ to a text generation task with input space $\mathcal{X} = \mathcal{V}^m$ and output space $\mathcal{Y} = \mathcal{V}^n$ modeled by $\pi(y|x) = \pi(x, y)/\pi(x)$.

A preference dataset $\mathcal{D}^{\text{pref}}$ consists of pairs of responses as the preference candidates, and their corresponding preferences pre-annotated by humans (Dubey et al., 2024) or strong LMs through prompting-based techniques (Dubois et al., 2024a). Then, a reward model $r_\phi : \mathcal{X} \times \mathcal{Y} \to \mathbb{R}$ is learned on $\mathcal{D}^{\text{pref}}$ and trained by minimizing the pair-wise preference loss by its general form:

$$\mathcal{L}(r_\phi) = \mathbb{E}_{(x,y_w,y_l)\sim\mathcal{D}^{\text{pref}}}[\ell(r_\phi(x, y_w) - r_\phi(x, y_l))], \tag{1}$$

where $y_w, y_l$ are the chosen and rejected preference candidates, and $\ell$ is a function that maps the difference between the two rewards into a probability; or its specific form:

$$\mathcal{L}(r_\phi) = \mathbb{E}_{(x,y_w,y_l)\sim\mathcal{D}^{\text{pref}}}\left[-\log\frac{e^{r_\phi(x,y_w)}}{e^{r_\phi(x,y_w)} + e^{r_\phi(x,y_l)}}\right], \tag{2}$$

where the preference is discretized, i.e., the chosen response $y_w$ is always annotated as better than the rejected response $y_l$ among different annotators, and the preference formulation is based on Bradley-Terry (BT) model definition.

Finally, a policy $\pi_\theta$ is learned to maximize the following alignment objective (Ziegler et al., 2019; Ji et al., 2024)

$$\mathcal{L}(\pi_\theta) = \mathbb{E}_{x\sim\mathcal{D}}(\mathbb{E}_{y\sim\pi_\theta(\cdot|x)}[r_\phi(x, y)] - \beta\mathbb{D}_{\text{KL}}[\pi_\theta(y|x)||\pi_{\text{ref}}(y|x)]), \tag{3}$$

where $\mathcal{D}$ is a task-specific dataset, $\pi_{\text{ref}}$ is the reference model, which is usually the initial checkpoint of $\pi_\theta$, typically a model supervised-finetuned (SFT-ed) on instruction-following datasets. $\mathbb{D}_{\text{KL}}$ is the Kullback-Leibler divergence loss and $\beta$ is a density coefficient.

### 3.2 RL FINE-TUNING

One standard approach to optimize the alignment objective Eq. (3) is to use RL algorithms, which is a consequence of the discrete nature of language generation. Recently, Ziegler et al. (2019) proposed to search for $\pi_\theta$ that maximizes a KL-regularized reward $r_\phi(x, y) - \beta\log\frac{\pi_\theta(y|x)}{\pi_{\text{ref}}(y|x)}$, which can be achieved by policy gradient methods, such as Proximal Policy Optimization (PPO, Schulman et al. (2017)) and Group Relative Policy Optimization (GRPO, Shao et al. (2024)).

### 3.3 DIRECT PREFERENCE OPTIMIZATION

Rafailov et al. (2023) proposed DPO that optimizes $\pi_\theta$ directly from the preference data. Eq. (3) can be organized as

$$\min_{\pi_\theta} \mathbb{E}_{x\sim\mathcal{D}}[\text{KL}(\pi_\theta(y|x)||\pi^*(y|x)) - \log Z(x)], \tag{4}$$

where the function $Z(x)$ satisfies $Z(x) = \sum_y \pi_{\text{ref}}(y|x)\exp(\frac{1}{\beta}r_\phi(x, y))$, and the optimal solution $\pi^*$ satisfies $\pi^*(y|x) = \frac{1}{Z(x)}\pi_{\text{init}}(y|x)\exp(\frac{1}{\beta}r_\phi(x, y))$.

The optimal solution of Eq. (4) is obtained when $\text{KL}(\pi_\theta||\pi^*)$ is minimized. Let $\pi_\theta^*$ be the optimal solution of Eq. (4), then $\pi_\theta^*$ equals to $\pi^*$. The relationship between $r_\phi$ and $\pi_\theta$ can be further expressed as:

$$r_\phi(x, y) = \beta\log\frac{\pi_\theta^*(y|x)}{\pi_{\text{ref}}(y|x)} + \beta\log Z(x). \tag{5}$$

Then, they proposed to directly optimize the policy $\pi_\theta$ by replacing $\pi_\theta^*$ with $\pi_\theta$ and substituting the corresponding reward function into a pair-wise preference loss:

$$\mathcal{L}_{\text{DPO}}(\pi_\theta) = \mathbb{E}_{(x,y_w,y_l)\sim\mathcal{D}^{\text{pref}}} \left[ -\log\sigma\left(\beta\log\frac{\pi_\theta(y_w|x)}{\pi_{\text{ref}}(y_w|x)} - \beta\log\frac{\pi_\theta(y_l|x)}{\pi_{\text{ref}}(y_l|x)}\right)\right]. \quad (6)$$

Our goal is to understand the requirements of preference candidates during the alignment process when performing alignment methods like DPO. In the following sections, we try to achieve our goal by answering the following two research sub-questions (**RQs**) empirically and theoretically:

**RQ1:**  Can we perform a qualitative description of the alignment process, or can we characterize the requirements of preference candidates through the alignment process?

**RQ2:**  Is it possible to ensure that the qualitative description of the alignment process has actionable insight and can help conduct the effective alignment approach?

## 4  EMPIRICAL ANALYSIS

### 4.1  ANALYSIS SETUP

**Models.**  We use different models including Llama-3-8B-Instruct (AI@Meta, 2024), Zephyr-sft-full (Tunstall et al., 2023) and Phi-2 (Li et al., 2023) for experiments. We select these models based on their parameter scales and training stages. We use PairRM (Jiang et al., 2023b) as the ground-truth preference model in our experiments, acting as a surrogate to expensive human preference for preference annotation. More details are shown in Appendix C.1.

**Dataset.**  We use the prompts and preference candidates from UltraFeedback (Cui et al., 2023), then relabeled the preference by PairRM to get the final off-policy dataset, aiming at ensuring the identical preference between different preference datasets. More details are shown in Appendix C.2.

**Benchmarks.**  Following previous works (Meng et al., 2024; Ji et al., 2024), We use AlpacaEval 2.0 (Dubois et al., 2024b) as our evaluation benchmark and report the length-controlled win rate over the reference responses. More details are shown in Appendix C.3.

### 4.2  MAIN RESULTS: THE EFFECTIVENESS DISCREPANCY BETWEEN OFF-POLICY/ON-POLICY DATA EXISTS

Firstly, we propose a two-iteration training framework for each model, incorporating a full combination of off-policy and on-policy candidates. For each model, we conduct four distinct training configurations: 1) $\text{PC}_{\text{off}\to\text{off}}$: Two consecutive iterations using off-policy candidates; 2) $\text{PC}_{\text{off}\to\text{on}}$: First iteration with off-policy candidates followed by on-policy candidates; 3) $\text{PC}_{\text{on}\to\text{off}}$: First iteration with on-policy candidates followed by off-policy candidates; and 4) $\text{PC}_{\text{on}\to\text{on}}$: Two iterations exclusively using on-policy candidates. We provide more details in Appendix C.4.

We present our result in Table 1. Our observation and conclusions are as follows. **1) The effectiveness discrepancy between $\text{PC}_{\text{off}}$ and $\text{PC}_{\text{on}}$ exists among different models.** For Llama-3, models trained with $\text{PC}_{\text{on}}$ consistently outperform those trained with $\text{PC}_{\text{off}}$ given the same initial model in every setting ($\Delta < 1$), which suggests $\text{PC}_{\text{on}}$ generally improve Llama-3 better than $\text{PC}_{\text{off}}$. However, results on Zephyr are observed to be different from those of Llama-3. Models trained with $\text{PC}_{\text{on}}$ outperform those with $\text{PC}_{\text{off}}$ when the initial model has been trained with $\text{PC}_{\text{off}}$ in the previous iteration ($\Delta > 1$). In other cases, $\text{PC}_{\text{on}}$ leads to a worse performance for Zephyr compared with $\text{PC}_{\text{off}}$ ($\Delta < 1$). For Phi-2, the results are opposite to those of Llama-3. Model trained with $\text{PC}_{\text{off}}$ consistently outperforms that with $\text{PC}_{\text{on}}$ in all settings ($\Delta > 1$). **2) The alignment process may result in a failure when using $\text{PC}_{\text{on}}$.** We observe a slight performance drop for Phi-2 when trained with $\text{PC}_{\text{on}}$, particularly if the initial model is the SFT model or has been trained with $\text{PC}_{\text{off}}$ in the previous iteration. **3) The effectiveness of $\text{PC}_{\text{off}}$ varies within the same model under different circumstances.** We observe varying improvements when optimizing Zephyr by $\text{PC}_{\text{off}}$ across different training iterations (12.7/3.0/8.5-point increase). The discrepancy between $\text{PC}_{\text{off}}$ and $\text{PC}_{\text{on}}$

| Iter-1 | Iter-2 | LC Win Rate | Win Rate | Avg. Len | $\Delta(\times)$ | LC Win Rate | Win Rate | Avg. Len | $\Delta(\times)$ | LC Win Rate | Win Rate | Avg. Len | $\Delta(\times)$ |
|---|---|---|---|---|---|---|---|---|---|---|---|---|---|
| | | **Llama-3-8B-Instruct** | | | | **Zephyr-7B** | | | | **Phi-2-2.7B** | | | |
| - | - | 24.59 | 24.47 | 1924 | - | 8.12 | 4.25 | 824 | - | 5.81 | 3.72 | 915 | - |
| $PC_{off}$ | - | $27.73_{(+3.14)}$ | 22.85 | 1605 | 0.33 | $20.77_{(+12.65)}$ | 19.99 | 1903 | 2.27 | $5.97_{(+0.16)}$ | 3.92 | 983 | $+\infty$ |
| $PC_{on}$ | - | $34.04_{(+9.45)}$ | 34.47 | 2014 | | $13.70_{(+5.58)}$ | 9.90 | 1278 | | $4.21_{(-1.60)}$ | 2.86 | 961 | |
| $PC_{off}$ | $PC_{off}$ | $27.83_{(+0.10)}$ | 24.38 | 1723 | <0.01 | $23.77_{(+3.00)}$ | 21.67 | 1757 | 0.24 | $6.44_{(+0.47)}$ | 4.43 | 1077 | $+\infty$ |
| $PC_{off}$ | $PC_{on}$ | $40.57_{(+12.84)}$ | 41.89 | 2094 | | $33.28_{(+12.51)}$ | 36.85 | 2575 | | $4.92_{(-1.05)}$ | 3.46 | 995 | |
| $PC_{on}$ | $PC_{off}$ | $36.36_{(+2.32)}$ | 36.58 | 2010 | 0.22 | $22.22_{(+8.52)}$ | 19.33 | 1656 | 1.56 | $5.73_{(+1.52)}$ | 3.77 | 991 | 1.13 |
| $PC_{on}$ | $PC_{on}$ | $44.52_{(+10.48)}$ | 50.57 | 2473 | | $19.16_{(+5.46)}$ | 18.05 | 1746 | | $5.55_{(+1.34)}$ | 3.68 | 946 | |

Table 1: Results of full-combination two-iteration experiments for all three models. "$PC_{on}$" and "$PC_{off}$" refer to on-policy and off-policy preference candidates respectively, "iter" is the abbreviation of "iteration". As focusing on the length-controlled win rate (LC Win Rate) of the benchmark, the red number shows the relative increase compared to the initial model (i.e., iter-2 compared to iter-1, iter-1 compared to SFT) while the green number shows the relative decrease. $\Delta$ shows the ratio relationship of relative increase between models trained with $PC_{off}$ and $PC_{on}$. "$+\infty$" means there is a performance drop when training on $PC_{off}$ or $PC_{on}$.

shows that during the alignment process, the requirements of preference candidates are dynamic. This patterned dynamic nature motivates our central proposal: the alignment stage assumption.

We introduce the **alignment stage assumption** to model the dynamic requirements of preference candidates. Specially, the alignment process can be divided into two stages, the preference injection stage and the preference fine-tuning stage. During the preference injection stage, $PC_{off}$ will be more effective; when the model comes into the preference fine-tuning stage, $PC_{off}$ will be less effective than $PC_{on}$. According to the results in Table 1 and the alignment stage assumption, we note that Llama-3 has been in the preference fine-tuning stage in all settings; after training on $PC_{off}$, Zephyr is in the preference fine-tuning stage; Phi-2 is in the preference injection stage in all settings.

## 4.3 DIVERSITY AND QUALITY AS THE CHARACTERISTICS OF ALIGNMENT STAGES (RQ1)

To answer **RQ1**, following previous works (Ding et al., 2024; Grillotti et al., 2024), we focus on the two key characteristics of preference data: intra-diversity and answer quality, and perform experiments on Zephyr. We use Zephyr since it shifts from the preference injection stage to the preference fine-tuning stage after training with $PC_{off}$. To de-confound the effects of data characteristics from their on-policy/off-policy nature, we introduce $PC_{llama}$, a dataset constructed off-policy with regard to Zephyr by sampling from Llama-3-8B-Instruct, then annotating preferences using PairRM. All prompts of $PC_{llama}$ are the same as $PC_{on}$ and $PC_{off}$. We provide more details in Appendix C.5.

$PC_{llama}$ is designed to isolate the impact of data characteristics. Through experiments, we show that the preference candidates in $PC_{off}$ have a higher intra-diversity than those in $PC_{llama}$, and quality of preference candidates in $PC_{off}$ is lower than that in $PC_{llama}$. We provide experimental details about the comparison between $PC_{off}$ and $PC_{llama}$ in Appendix C.5. Besides results of models trained with $PC_{off}$ and $PC_{llama}$, we also include the $PC_{on}$ results as references.

We present our results in Table 2. Our observations and conclusions are as follows. **1) High diversity is more effective for models in the preference injection stage.** Compared with the SFT baseline, model trained with $PC_{off}$ achieves a 12.7-point performance increase. In contrast, model trained with $PC_{llama}$ achieves a 5.4-point performance increase, which is similar to the model trained with $PC_{on}$ that achieves a 5.6-point performance increase. However, when Zephyr has been in the preference fine-tuning stage, $PC_{off}$ achieves a relatively smaller performance increase, which is 3.0 points, compared

Table 2: Results of Zephyr-7B for RQ1.

| Iter-1 | Iter-2 | LC Win Rate | Win Rate |
|---|---|---|---|
| - | - | 8.12 | 4.25 |
| $PC_{off}$ | - | $20.77_{(+12.65)}$ | 19.99 |
| $PC_{llama}$ | - | $13.53_{(+5.41)}$ | 10.15 |
| $PC_{on}$ | - | $13.70_{(+5.58)}$ | 9.90 |
| $PC_{off}$ | $PC_{off}$ | $23.77_{(+3.00)}$ | 21.67 |
| $PC_{off}$ | $PC_{llama}$ | $29.32_{(+8.55)}$ | 37.03 |
| $PC_{off}$ | $PC_{on}$ | $33.28_{(+12.51)}$ | 36.85 |

with $PC_{llama}$ and $PC_{on}$, which are 8.6 points and 12.5 points, respectively. Similar results are also observed from experiments in § 4.2, where $PC_{off}$ attributes to slight improvement for Llama-3. **2) High quality will be more effective for models in the preference fine-tuning stage.** For the model in the preference fine-tuning stage, being trained with $PC_{llama}$ achieves a 8.6-point increase. However, the relative performance increase is only 5.4 points when trained with $PC_{llama}$ for model

in the preference injection stage. As $PC_{llama}$ being a dataset with off-policy preference candidates with regard to Zephyr-7B, the dynamic effectiveness is attributed to the dynamic requirements for models in different stages, where we conclude that quality matters at the second stage.

The narrative explanation of different stage characteristics is through dynamic alignment goals. Model in the preference injection stage performs poorly and lacks knowledge about ground-truth preference and its corresponding high-reward region. The exploration will be low-effective since the high-reward region can hardly be explored. Data with high diversity aims at injecting preference knowledge into policy models. For the models in the preference fine-tuning stage, it is low-effective to perform large-scale preference injection, and the alignment goal shifts to explore high-reward region, sampling responses that are of high quality.

## 5 BOUNDARY MEASUREMENT ALGORITHM THAT DETERMINES THE BOUNDARY BETWEEN ALIGNMENT STAGES (RQ2)

In this section, we analyze the requirements of preference data from a theoretical perspective. We show the equivalence between the DPO objective and the alignment optimization objective (§5.1) and conclude that we are finding a better text distribution estimation to general text distribution defined by ground-truth preference model when choosing preference candidates (§5.1). To find a better text distribution, we introduce the preference consistency, which is the sufficient condition of identical distributions between some text distribution $\pi$ and general text distribution $\pi_G$ (§5.2). Finally, we propose the **boundary measurement algorithm**, a practical estimation of the preference consistency measurement (§5.3). We illustrate the relationship between text distribution estimation and boundary measurement algorithm in Figure 4, Appendix D. All proofs are shown in Appendix E.

**Notation.** Generally, let $\pi$ be a policy that represents a text distribution. Following the notation in §3.1, let $\mathbb{P} : \mathcal{X} \times \mathcal{Y} \times \mathcal{Y} \rightarrow [0, 1]$ be the preference distribution that satisfies Bradley-Terry definition with respect to reward model $r$. The output $\mathbb{P}(y_1 \succ y_2|x)$ represents the preference of $y_1$ outperforming $y_2$. Specifically, let $\pi_G$ be the general policy and the general text distribution, $\pi_{off}$ be an abstract policy that generates the candidates of $PC_{off}$, $\pi_\theta$ be the policy that generates the preference candidates of $PC_{on}$, $\pi^*$ be an optimal solution of $\pi$. $\mathbb{P}^*$ is the ground-truth preference distribution derived from the ground-truth reward model $r^*$. $\mathbb{P}_\theta$ is the parameterized preference distribution derived from $r_\phi$, which is the analytical solution of Eq. (5) given $\pi_\theta$ and $\pi_{ref}$.

### 5.1 OPTIMIZATION CONSISTENCY ANALYSIS

Eq. (5) establishes a one-way mapping between the reward model and policy model that for every reward model $r_\phi$, there exists a policy $\pi_\theta^*$ that satisfies Eq. (5) and $\pi_\theta^*$ is the optimal solution of Eq. (3). First of all, we show that the one-way mapping is reversible, i.e., Eq. (5) satisfies for every $\pi_\theta$ when optimizing through Eq. (6).

**Theorem 5.1.** (Bijection between reward function and policy) *Under mild assumption, for any policy $\pi_\theta$ and the static reference model $\pi_{ref}$, there exists a unique reward model $r_\phi$ satisfying $\pi_\theta$ being the optimal solution of Eq. (3).*

Theorem 5.1 indicates that the optimization objective of Eq. (6) and the alignment objective Eq. (3) are theoretically equivalent. We then discuss the condition that achieves the optimal solution of Eq. (3) via Eq. (6).

**Theorem 5.2.** *(*The Necessary condition of the optimal solution of Eq. (3)) *The optimal solution of Eq. (3) can only be achieved if the preference dataset $\mathcal{D}^{pref}$ has infinite preference data.*

Theorem 5.2 indicates that **1) The optimal solution of the general alignment objective is practically intractable,** as it is impossible to construct a preference dataset with infinite preference candidates. Given limited preference candidates, the optimization objective is the preference consistency between $\mathbb{P}^*$ and $\mathbb{P}_\theta$ within the limited dataset. **2) The alignment process will be more effective if the limited preference dataset is a well-defined proxy of the infinite-sample preference dataset.** Assuming that the preference candidates, i.e., text-based responses of the infinite-sample preference dataset, are sampled from the general text distribution, then we are estimating general text distribution when selecting preference candidates.

## 5.2 THE GENERAL TEXT DISTRIBUTION ESTIMATION

In this section, we aim at finding a measurement that can estimate the distance between the general text distribution $\pi_G$ and the parameterized text distribution $\pi_\theta$. Regular distance measurement like KL divergence does not work since both text distributions are intractable. Instead, we aim to measure the consistency of the preference distributions between $\mathbb{P}^*$ and $\mathbb{P}_\theta$, which we will show to be a sufficient condition of $\pi_G$ and $\pi_\theta$ being identical. First of all, we formally introduce the definition of $\pi_G$ and $\mathbb{P}_\theta$ in Definition 5.3.

**Definition 5.3.** The general text distribution $\pi_G$ is defined by the ground-truth $\mathbb{P}^*$ that satisfies

$$\mathbb{P}^*(y_1 \succ y_2 | x) = \sigma(\log \pi_G(y_1 | x) - \log \pi_G(y_2 | x)), \tag{7}$$

and the parameterized preference given $\pi_\theta$ is defined as

$$\mathbb{P}_\theta(y_1 \succ y_2 | x) = \sigma(\log \pi_\theta(y_1 | x) - \log \pi_\theta(y_2 | x)). \tag{8}$$

We note that Definition 5.3 is not related with the optimal condition defined in Eq. (3) and Eq. (5). That is because we will not introduce any assumptions premised on optimizing Eq. (3), and the general text distribution should be irrelevant to hyper-parameter $\beta$ and reference model $\pi_{\text{ref}}$.

**Theorem 5.4.** (The uniqueness of $\pi_G$) *There exists a unique $\pi_G$ under Definition 5.3 given a well-defined $\mathbb{P}^*$.*

Theorem 5.4 and Definition 5.3 indicate that $\mathbb{P}^*$ and $\pi_G$ form a pair of bijections, which allows us to estimate $\pi_G$ by estimating $\mathbb{P}^*$. We can thus measure the distance between two preference distributions that are derived from $\pi_G$ and $\pi_\theta$ respectively as a proxy of the estimation between text distributions. First of all, we provide the definition of preference consistency in Definition 5.5.

**Definition 5.5.** Given preference distribution $\mathbb{P}_1$ and $\mathbb{P}_2$ based on BT definition, the consistency between $\mathbb{P}_1$ and $\mathbb{P}_2$ is defined by the following formula:

$$\mathbb{E}_{x,y_1,y_2}\left[\mathbb{I}\left[\mathbb{P}_1(y_1 \succ y_2 | x)\right] \odot \mathbb{I}\left[\mathbb{P}_2(y_1 \succ y_2 | x)\right]\right] \tag{9}$$

where $\mathbb{I} : [0,1] \to \{0,1\}$ is the indicator function that maps values in the interval $[0, 0.5]$ into 0 and values in $(0.5, 1]$ into 1. $\odot$ is the XNOR operator.

The preference consistency defined in Definition 5.5 achieves its maximum when $\mathbb{I}\left[\mathbb{P}_1(y_1 \succ y_2 | x)\right] = \mathbb{I}\left[\mathbb{P}_2(y_1 \succ y_2 | x)\right]$ satisfies for any $\{x, y_1, y_2\}$, which is a sufficient condition of two identical preference distributions. In other words, preference consistency is to determine if probabilities of identical samples exhibit identical rank orders for both text distributions.

## 5.3 PRACTICAL ESTIMATION OF PREFERENCE CONSISTENCY

Given on-policy distribution $\pi_\theta$ and off-policy distribution $\pi_{\text{off}}$, we perform the preference consistency measurement between these distributions and the general text distribution $\pi_G$. Let $\{y_1^i\}_m, \{y_2^i\}_n$ be the responses sampled from $\pi_\theta$ and $\pi_{\text{off}}$ given prompt $x$ with size $m$ and $n$, respectively. For each prompt $x$, We estimate the preference consistency by responses sampled from both $\pi_\theta$ and $\pi_{\text{off}}$ to reduce sampling variance:

$$\frac{1}{mn}\sum_{y_1^i}^{m}\sum_{y_2^j}^{n}\mathbb{I}\left[\mathbb{P}^*(y_1^i \succ y_2^j | x)\right] \odot \mathbb{I}\left[\mathbb{P}_\theta(y_1^i \succ y_2^j | x)\right], \tag{10}$$

which measures the consistency between $\mathbb{P}^*$ and $\mathbb{P}_\theta$, and

$$\frac{1}{mn}\sum_{y_1^i}^{m}\sum_{y_2^j}^{n}\mathbb{I}\left[\mathbb{P}^*(y_1^i \succ y_2^j | x)\right] \odot \mathbb{I}\left[\mathbb{P}_{\text{off}}(y_1^i \succ y_2^j | x)\right], \tag{11}$$

which measures the consistency between $\mathbb{P}^*$ and $\mathbb{P}_{\text{off}}$. Practically, we assume that $\pi_\theta$ and $\pi_{\text{off}}$ are highly divergent text distributions and responses are sampled from largely distinct regions of the vast text space, which allows that $\mathbb{I}[\mathbb{P}_\theta(y_1^i \succ y_2^j | x)] = 1$ and $\mathbb{I}[\mathbb{P}_{\text{off}}(y_1^i \succ y_2^j | x)] = 0$, an assumption empirically supported in Appendix F.1. This allows the preference consistency between $\mathbb{P}^*$ and $\mathbb{P}_\theta, \mathbb{P}_{\text{off}}$ to be simplified into $\frac{1}{mn}\sum_{y_1^i}^{m}\sum_{y_2^j}^{n}\mathbb{I}\left[\mathbb{P}^*(y_1^i \succ y_2^j | x)\right]$ and $\frac{1}{mn}\sum_{y_1^i}^{m}\sum_{y_2^j}^{n}\mathbb{I}\left[\mathbb{P}^*(y_2^j \succ y_1^i | x)\right]$,

| Iter-1 | Iter-2 | LC Win Rate | Win Rate | BS (initial) | $\Delta(\times)$ | LC Win Rate | Win Rate | BS (initial) | $\Delta(\times)$ | LC Win Rate | Win Rate | BS (initial) | $\Delta(\times)$ |
|---|---|---|---|---|---|---|---|---|---|---|---|---|---|
| | | Llama-3-8B-Instruct | | | | Zephyr-7B | | | | Phi-2-2.7B | | | |
| - | - | 24.59 | 24.47 | - | - | 8.12 | 4.25 | - | - | 5.81 | 3.72 | - | - |
| $\text{PC}_{\text{off}}$ | - | $27.73_{(+3.14)}$ | 22.85 | 0.62 | 0.33 | $20.77_{(+12.65)}$ | 19.99 | 0.40 | 2.27 | $5.97_{(+0.16)}$ | 3.92 | 0.23 | $+\infty$ |
| $\text{PC}_{\text{on}}$ | - | $34.04_{(+9.45)}$ | 34.47 | | | $13.70_{(+5.58)}$ | 9.90 | | | $4.21_{(-1.60)}$ | 2.86 | | |
| $\text{PC}_{\text{off}}$ | $\text{PC}_{\text{off}}$ | $27.83_{(+0.10)}$ | 24.38 | 0.66 | $<0.01$ | $23.77_{(+3.00)}$ | 21.67 | 0.66 | 0.24 | $6.44_{(+0.47)}$ | 4.43 | 0.25 | $+\infty$ |
| $\text{PC}_{\text{off}}$ | $\text{PC}_{\text{on}}$ | $40.57_{(+12.84)}$ | 41.89 | | | $33.28_{(+12.51)}$ | 36.85 | | | $4.92_{(-1.05)}$ | 3.46 | | |
| $\text{PC}_{\text{on}}$ | $\text{PC}_{\text{off}}$ | $36.36_{(+2.32)}$ | 36.58 | 0.69 | 0.22 | $22.22_{(+8.52)}$ | 19.33 | 0.58 | 1.56 | $5.73_{(+1.52)}$ | 3.77 | 0.23 | 1.13 |
| $\text{PC}_{\text{on}}$ | $\text{PC}_{\text{on}}$ | $44.52_{(+10.48)}$ | 50.57 | | | $19.16_{(+5.46)}$ | 18.05 | | | $5.55_{(+1.34)}$ | 3.68 | | |

Table 3: Results of full-combination two-iteration experiments. The "BS (initial)" denotes the relative boundary score of each initial policy, calculated as $V_{\text{off}}/(V_{\text{off}} + V_{\text{on}})$ from the results of the boundary measurement algorithm shown in Algorithm 1. A score less than 0.5 indicates the policy in the preference injection stage and thus dataset with better intra-diversity will be more efficient ($\Delta{>}1$). Otherwise, it is in preference fine-tuning stage and thus the quality matters ($\Delta{<}1$).

respectively. Under mild assumptions, these equations indicate that it is possible to select a better proxy of $\pi_G$ from $\pi_\theta$ and $\pi_{\text{off}}$ by comparing preference consistency of $\pi_\theta$ and $\pi_{\text{off}}$ regarding to $\mathbb{P}^*$.

---

**Algorithm 1** Boundary Measurement Algorithm

---

1: **Input** Preference datasets $\text{PC}_{\text{on}}$, $\text{PC}_{\text{off}}$, Preference model $\mathbb{P}$.
2: $V_{\text{on}}, V_{\text{off}} \leftarrow 0, 0$
3: **for** $(x, y_1, y_2) \sim \text{PC}_{\text{on}}$ **do**
4:     Sample the paired responses $(y'_1, y'_2)$ from $\text{PC}_{\text{off}}$ where the input of paired responses $x'$ is equal to $x$.
5:     **for** $y, y'$ where $y \in \{y_1, y_2\}, y' \in \{y'_1, y'_2\}$ **do**
6:         Update $V_{\text{on}} \leftarrow V_{\text{on}} + 1$ if $\mathbb{P}$ prefers $y$ better than $y'$ given $x$. Otherwise, update $V_{\text{off}}$.
7:     **end for**
8: **end for**
9: **if** $V_{\text{off}}{>}V_{\text{on}}$ **then**
10:     **return** Model is in the preference injection stage, $\text{PC}_{\text{off}}$.
11: **else**
12:     **return** Model is in the preference fine-tuning stage, $\text{PC}_{\text{on}}$.
13: **end if**

---

We then provide the boundary measurement algorithm in Algorithm 1, which is the preference consistency measurement when letting $m = n = 2$. The algorithm shows that alignment stages are decided by preference dataset and preference model jointly. In other words, one initial policy can be in preference injection stage and preference fine-tuning stage at the same time given different off-policy preference candidates and preference models. However, once the preference model and off-policy preference dataset are given, we can decide the alignment stage that model is currently in, and thus optimizing preference data for policy models.

## 5.4 EXPERIMENTS OF THE BOUNDARY MEASUREMENT ALGORITHM

Following the experiment settings in §4, we perform experiments on three base models to verify the effectiveness of the boundary measurement algorithm. We present our result in Table 3. For Llama-3, the results fit the stage assumption well. The boundary scores are greater than 0.5 for all initial models, indicating that Llama-3 is in preference fine-tuning stage. The results for Phi-2 also align with the stage assumption, as the boundary scores are less than 0.5 for all initial models, showing that the model is in preference injection stage. For Zephyr, the results fit the assumption well given the SFT model or the model trained with $\text{PC}_{\text{off}}$ as the initial models. We notice that the model trained with $\text{PC}_{\text{on}}$ has a positive score (0.58), but the follow-up training with $\text{PC}_{\text{off}}$ (an 8.5-point increase) is still more effective than $\text{PC}_{\text{on}}$ (a 5.5-point increase). We attribute it to the lower quality of $\text{PC}_{\text{on}}$ relative to $\text{PC}_{\text{off}}$. We measure the quality of $\text{PC}_{\text{on}}$ following the comparison method used in Appendix C.5. The result shows that that the length-controlled win rate of $\text{PC}_{\text{on}}$ compared with $\text{PC}_{\text{off}}$ is 0.46, indicating that the quality of $\text{PC}_{\text{on}}$ is lower than that of $\text{PC}_{\text{off}}$.

## 6 GENERALIZABILITY ANALYSIS

In this section, we further extend the experiments on two models (Qwen2.5-1.5B (Yang et al., 2024) and Pythia-6.9B (Biderman et al., 2023)) and another LM alignment method (SLiC-HF (Zhao et al., 2023)) to verify the generalizability of our conclusions.

| Iter-1 | Iter-2 | LC Win Rate | Win Rate | BS (initial) | $\Delta(\times)$ | LC Win Rate | Win Rate | BS (initial) | $\Delta(\times)$ |
|---|---|---|---|---|---|---|---|---|---|
| | | Qwen2.5-1.5B | | | | Pythia-6.9B | | | |
| - | - | 5.41 | 3.00 | - | - | 1.81 | 1.06 | - | - |
| $PC_{off}$ | - | $7.24_{(+1.83)}$ | 8.78 | 0.35 | $+\infty$ | $1.28_{(-0.53)}$ | 2.45 | 0.22 | - |
| $PC_{on}$ | - | $4.85_{(-0.56)}$ | 2.69 | | | $1.02_{(-0.79)}$ | 1.48 | | |
| $PC_{off}$ | $PC_{off}$ | $9.27_{(+3.86)}$ | 10.06 | 0.47 | 9.41 | $2.51_{(+1.23)}$ | 4.72 | 0.26 | 1.68 |
| $PC_{off}$ | $PC_{on}$ | $7.65_{(+0.41)}$ | 11.12 | | | $2.01_{(+0.73)}$ | 3.25 | | |
| $PC_{on}$ | $PC_{off}$ | $7.08_{(+2.23)}$ | 8.58 | 0.38 | 2.48 | $2.79_{(+1.77)}$ | 3.46 | 0.24 | 1.49 |
| $PC_{on}$ | $PC_{on}$ | $5.75_{(+0.90)}$ | 3.45 | | | $2.21_{(+1.19)}$ | 3.12 | | |

Table 4: Results of two-iteration experiments in Qwen2.5-1.5B and Pythia-6.9B.

**Generalizability Analysis on Additional LMs.** We further extend experiments on Qwen2.5-1.5B (Yang et al., 2024) and Pythia-6.9B (Biderman et al., 2023). We follow the experiment settings in §4 and train the models on UltraChat for one epoch first. We report the results in Table 4. The results show that the effectiveness discrepancy between $PC_{on}$ and $PC_{off}$ exists. Specifically, the boundary score shows that the initial checkpoints of the two models, i.e., the SFT checkpoint and the checkpoints trained on $PC_{on}$ and $PC_{off}$ in the first iteration are all in the preference injection stage. As shown in the results, the performance of models trained on $PC_{off}$ outperforms those trained on $PC_{on}$ given the same initial checkpoint among different models, which fit the our conclusions well.

**Generalizability Analysis on SLiC-HF.** Though the empirical analysis of the two-stage assumption and the theoretical analysis of the boundary measurement are based on DPO, we show that the assumption and our conclusions can be further extended to other LM alignment methods. In this section, We perform experiments on SLiC-HF (Zhao et al., 2023).

We report the result in Table 5. The results show a similar trend as those aligning with DPO, where we observe the effectiveness discrepancy between $PC_{on}$ and $PC_{off}$ for different models. By performing the alignment stage assumption for these models and performing the boundary measurement, we observe a similar result as those aligning with DPO, which shows that the effectiveness discrepancy exists, and we can apply the two-stage assumption and judge the boundary between stages via the boundary measurement we proposed in Algorithm 1.

| Iter-1 | Iter-2 | LC Win Rate | Win Rate | BS (initial) | $\Delta(\times)$ | LC Win Rate | Win Rate | BS (initial) | $\Delta(\times)$ | LC Win Rate | Win Rate | BS (initial) | $\Delta(\times)$ |
|---|---|---|---|---|---|---|---|---|---|---|---|---|---|
| | | Llama-3-8B-Instruct | | | | Zephyr-7B | | | | Phi-2-2.7B | | | |
| - | - | 24.59 | 24.47 | - | - | 8.12 | 4.25 | - | - | 5.81 | 3.72 | - | - |
| $PC_{off}$ | - | $28.88_{(+4.38)}$ | 27.51 | 0.62 | 0.68 | $17.73_{(+9.61)}$ | 16.94 | 0.40 | 1.35 | $5.97_{(+0.16)}$ | 4.68 | 0.23 | $+\infty$ |
| $PC_{on}$ | - | $31.06_{(+6.47)}$ | 39.68 | | | $15.26_{(+7.14)}$ | 10.44 | | | $5.32_{(-0.49)}$ | 4.32 | | |
| $PC_{off}$ | $PC_{off}$ | $28.18_{(-0.70)}$ | 23.71 | 0.66 | - | $21.59_{(+3.86)}$ | 20.18 | 0.65 | 0.38 | $8.55_{(+2.58)}$ | 9.64 | 0.40 | 1.43 |
| $PC_{off}$ | $PC_{on}$ | $12.66_{(-11.93)}$ | 5.12 | | | $25.32_{(+7.59)}$ | 28.81 | | | $7.77_{(+1.80)}$ | 6.11 | | |
| $PC_{on}$ | $PC_{off}$ | $32.63_{(+1.57)}$ | 30.38 | 0.71 | 0.19 | $19.84_{(+4.58)}$ | 15.18 | 0.60 | 0.98 | $6.38_{(+1.06)}$ | 5.83 | 0.35 | 1.54 |
| $PC_{on}$ | $PC_{on}$ | $39.46_{(+8.40)}$ | 51.67 | | | $19.93_{(+4.67)}$ | 17.70 | | | $6.01_{(+0.69)}$ | 3.63 | | |

Table 5: Results of full-combination two-iteration experiments performed with SLiC-HF loss. The boundary score can be a good measurement to decide the boundary between each alignment stage.

Though the result matches the assumption and algorithm in most cases, we also observe a model collapse phenomenon for Llama-3 trained with $PC_{off}$ and $PC_{on}$ subsequently, where a very serious performance degradation is observed. It may result in the difference between DPO and SLIC-HF, as a similar performance degradation is not observed when aligning with DPO as shown in Table 3.

# 7 CONCLUSION AND LIMITATION

In this work, we reveal the effectiveness discrepancy between on-policy data and off-policy data for different models, and propose the alignment stage assumption when performing LM alignment through DPO. We characterize each alignment stage through analyzing the discrepancy by diversity and quality. We provide the boundary measurement algorithm, a theoretical-grounded method to decide the alignment stages. Though being an effective simplified abstraction of alignment process, the alignment stage assumption inspires exploration of smoother and more adaptive data blending strategies rather than a rigid switch, which is not included and we leave for future research.

## REPRODUCIBILITY STATEMENT

To ensure the reproducibility of our work and to facilitate a clearer understanding of our contributions, we provide extensive supporting materials. In the main text, we describe the models, benchmark and training parameters we used in our experiment in §4. In Appendix C, we provide further detailed information, including model details, evaluation details, data details and training details. Our work is based on open-sourced models, open-sourced dataset and open-sourced benchmark, which ensures our results are reproducible.

## ACKNOWLEDGMENTS

We thank the insightful suggestions of anonymous reviewers. This work is supported by grants: National Natural Science Foundation of China (Grant No. 62406088 and 62422603), Guangdong Basic and Applied Basic Research Foundation (Grant No. 2025A1515011376), Guangdong Basic and Applied Basic Research Foundation (Grant No. 2024B0101050003)

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

## A   THE USE OF LARGE LANGUAGE MODELS

In this work, we utilized LLMs solely for the purpose of polishing writing. The LLMs were not used for content generation, and all research, analysis, and conclusions presented are the result of our own work and independent thought.

## B   FURTHER DISCUSSION

### B.1   COMPUTATIONAL COST OF ALGORITHM 1

The boundary measurement algorithm requires a one-time comparison on a subset of the data, which requires performing on-policy sampling by current policy to acquire $PC_{on}$. In our experiments, we use $2,000$ prompts in the "test_prefs" split of UltraFeedback (binarized) dataset for this measurement. Specifically, we compare the on-policy samples generated by current policy and the off-policy samples derived fom the "test_prefs" split of the UltraFeedback (binarized) dataset, using PairRM as the preference model. Compared to full DPO training on the $61,135$-sample dataset, the computational overhead of our boundary measurement is negligible, estimated to be 3.2% of a single training epoch. This demonstrates that our method is not only effective but also highly efficient and practical for real-world application.

### B.2   DEPENDENCY OF PREFERENCE MODEL

A key aspect of our boundary measurement is its reliance on a given preference model $\mathbb{P}$ to define the ground truth for the stage decision. This means the resulting stage boundary is relative to the preference model $\mathbb{P}$. If $\mathbb{P}$ is weak or biased, the boundary decision might be suboptimal for alignment towards true human preferences, but it will still be optimal for aligning towards the world view of $\mathbb{P}$. This highlights the importance of the choice of the preference model, a factor common to all preference-based alignment methods.

### B.3   CONNECTION WITH EXPLORATION-EXPLOITATION

Our two-stage assumption can be viewed as a simplified instantiation of the classic exploration-exploitation trade-off in reinforcement learning within the context of LM alignment. While traditional reinforcement learning focuses on exploration in state-action space, our work suggests that for LM alignment via preference-based alignment methods like DPO, exploration happens in the space of preference candidates. Choosing preference candidates with high diversity can be regarded as a form of exploration, where the model seeks to learn broadly about the reward landscape defined by preference model; while choosing high-quality preference candidates can be regarded as a form of exploitation, where the model refines its policy within high-reward regions defined by preference model. Our boundary measurement algorithm, therefore, acts as an adaptive switch between the exploration phrase and the exploitation phrase.

### B.4   DISCUSSION ABOUT DISTRIBUTION SHIFT THEORY

One possible confusion about the empirical analysis about stage characteristics we introduced in §4.3 lies in the contradiction between stage characteristics and distribution shift theory. Different from quantifying preference candidates by diversity and quality, $PC_{on}$ is an "in-domain" dataset, as its preference candidates are sampled from the current policy, while $PC_{off}$ is an "out-of-domain" dataset, as its preference candidates are sampled from models different from the current policy. As a consequence, the effectiveness of $PC_{on}$ may lie in its sharing the identical sampling distribution during the alignment process with regard to current policy. We alleviate the influence of distribution shift from two aspects.

First of all, the distribution shift theory posits that on-policy data is always superior to off-policy data. However, our results in §4.2 showing that optimizing models based on preference candidates sampled from their identical distribution is not always effective, which indicates that distribution shift is not the sole, or even the primary factor towards LM alignment. For example, for Phi-2, training with $PC_{on}$ leads to a performance drop, while training with $PC_{off}$, whose samples are from

a more distant distribution, leads to a performance increase. Secondly, we de-confound the effects of data characteristics (i.e., diversity and quality) from their on-policy/off-policy natures. Specifically, we use $PC_{\text{llama}}$ in §4.3, whose preference candidates are sampled from another model (i.e., Llama-3-8B-Instruct) that is distant to current policy (i.e., Zephyr-7B). Through empirical analysis about $PC_{\text{off}}$ and $PC_{\text{llama}}$ introduced in §C.5, we quantify the characteristics of $PC_{\text{off}}$ and $PC_{\text{llama}}$. This allows us to isolate the impact of data characteristics.

### B.5 DISCUSSION ABOUT IMPORTANCE OF PREFERENCE DATA SELECTION

As the field of LLMs matures beyond the primary pursuit of scale, the central challenges have shifted towards efficiency, reliability, and cost-effective customization. The decision of how to construct the effective dataset lies at the heart of this new paradigm. On-policy data generation, while providing highly relevant samples, introduces significant computational and financial overhead, acting as a major bottleneck for the widespread adoption and specialized fine-tuning of models. Our work addresses this challenge by moving the data selection process from an empirical art to a principled, stage-aware science. In an era increasingly focused on Data-Centric AI, instead of simply assuming on-policy data is superior, off-policy data can be more effective than on-policy data in some cases. By introducing the same LM inference overhead to construct the on-policy preference candidates and then optimizing LLMs, the model will achieve better performance when it has been in the preference fine-tuning stage. Our work provides a diagnostic framework to understand the model alignment stage and to strategically choose data that maximizes efficiency. Our research offers a critical methodology for building better-aligned models more efficiently and reliably, a core necessity for the next generation of AI systems.

### B.6 ADDITIONAL RELATED WORK ABOUT OPTIMIZATION VIA ON-POLICY AND OFF-POLICY CURRICULUM

On-policy reinforcement learning encourages LMs to perform active exploration during the optimization process, which enhances their generalization ability by learning from feedback on their own sampled outputs (Chen et al., 2025; Chu et al., 2025). However, on-policy sampling is expensive and time-consuming, and can result in policy degradation caused by entropy collapse or over-exploitation of sub-optimal responses (Yu et al., 2025). Relatively, optimizing LMs via off-policy data is cheap and stable, while it struggles with limited exploration and learning from novel, self-generated responses. To this end, several works focus on the optimizing LMs via both on-policy and off-policy data in an empirical and straight-forward way. LUFFY incorporates off-policy responses in Group Relative Policy Optimization method (GRPO, Shao et al. (2024)) by adding them directly to the group of on-policy responses (Yan et al., 2025). Qwen3 utilizes a weak-to-strong curriculum strategy during the optimization process, training models in off-policy responses and on-policy responses subsequently in a supervised fine-tuning manner to improve the reasoning capability of LMs (Yang et al., 2025). Other works aim at incorporating off-policy data and on-policy data in a sequential *SFT-then-RL* paradigm, either utilizing multi-task learning to balance SFT loss and RL objective at the same time (Zhang et al., 2025b) or performing SFT first, then RL (Lambert et al., 2024; Liu et al., 2025). In LM alignment scenario, our work reveals a patterned effectiveness discrepancy between off-policy data and on-policy data across different models, and proposes the alignment stage assumption to model the dynamic data requirements during the alignment process. Our work is aligned with findings in Reinforcement Learning from Verifiable Reward (RLVR) and offline RL sencarios, and can provide valuable and actionable discoveries for these fields.

## C TRAINING AND EVALUATION DETAILS

### C.1 MODEL DETAILS

Llama-3-8B-Instruct is a large language model with 8B parameter size, and has been aligned with human preferences for helpfulness and safety through supervised fine-tuning (SFT) and reinforcement learning from human feedback (RLHF). Zephyr-sft-full is a large language model with 7B parameter size, and is an aligned version of Mistral-7B (Jiang et al., 2023a) that has previously supervised fine-tuned on UltraChat (Ding et al., 2023) dataset. Phi-2 is a pretrained language model with 2.7B parameter size, and has not been fine-tuned or aligned on downstream tasks. Following

the setup process and training settings of Zephyr-sft-full, we conduct supervised fine-tuning on Phi-2 on UltraChat for one epoch to get the fine-tuned checkpoint for alignment experiments. These models vary on the model scale and training stage, which will result in different behavior in the subsequent experiments and be helpful for our analysis. We use PairRM (Jiang et al., 2023b) as the ground-truth preference model in our experiments, an efficient pair-wise preference model of size $0.4B$. PairRM is based on DeBERTA-V3 (He et al., 2023) and has been fine-tuned on high-quality preference datasets. Results on benchmarks like Auto-J Pairwise dataset (Li et al., 2024) show that PairRM outperforms most of the model-based reward models and performs comparably with larger reward models like UltraRM-13B (Cui et al., 2023). The reference model $\pi_{\text{ref}}$ we used in different experiment is the initial checkpoint of the corresponding policy model.

## C.2 DATASET DETAILS

UltraFeedback (Cui et al., 2023) is a large-scale, fine-grained, diverse preference dataset for LM alignment. UltraFeedback consists of $63,967$ prompts from diverse sources (including Ultra-Chat (Ding et al., 2023), ShareGPT (Chiang et al., 2023), Evol-Instruct (Xu et al., 2024), Truth-fulQA (Lin et al., 2022), FalseQA (Hu et al., 2023), and FLAN (Longpre et al., 2023)). For each prompt, the authors query multiple LLMs to generate 4 different responses, then the responses are scored and ranked by GPT-4 (OpenAI, 2023) based on criterion including instruction-following, truthfulness, honesty and helpfulness. To construct the UltraFeedback (binarized) dataset, the response with the highest overall score is selected as the "chosen" completion, and one of the remaining 3 responses at random as the "rejected" one, thus constructing the preference pairs.

We sample two answers by the current policy to acquire on-policy preference candidates. Specifically, we use all of the prompts derived from UltraFeedback, sample two responses as the preference candidates, then annotate the preference between the preference candidates by PairRM. We called "blender.compare_conversations" method to annotate the preference between preference candidates, which is the official method provided by the authors of PairRM. To ensure the consistency of preference annotators between off-policy preference dataset (whose preferences are annotated by GPT-4) and on-policy preference dataset (whose preferences are annotated by PairRM), We relabeled the preference of preference candidates in UltraFeedback (binarized) dataset by PairRM in the same way as labeling the preference in the on-policy preference dataset.

## C.3 EVALUATION DETAILS

AlpacaEval 2.0 (Dubois et al., 2024a) is a leading benchmark that assesses LLMs' instruction-following ability and alignment with human preference. To construct the AlpacaEval test set, the authors combine a variety of instruction-following datasets like self-instruct (Wang et al., 2023), open-assistant (Köpf et al., 2023), vicuna (Chiang et al., 2023), koala (Geng et al., 2023) and hh-rlhf (Bai et al., 2022), and finally construct a dataset with $805$ samples. It calculates the probability that an LLM-based evaluator (gpt-4-1106-preview) prefers the model output over the response generated by GPT-4, which provides an affordable and replicable alternative to human preference annotation. The win rate over the GPT-4 baseline is computed as the expected preference probability. The length-controlled win rate is a modified version that reduces the length bias, which alleviates reward hacking and prevents flawed judgment. We report the length-controlled win rate as it correlates best with Chatbot Arena (Dubois et al., 2024b), the real-world alignment benchmark based on human evaluation.

## C.4 EXPERIMENT DETAILS

For each training iterations, we use the initial checkpoint of current policy as the reference model. For on-policy experiments, we sample two answers from the current policy, using prompts same as UltraFeedback, then annotate the preference by PairRM. The hyper-parameters when training models are shown in Table 7. The hyper-parameters when generating on-policy preference candidates are shown in Table 6.

In practice, we seldom see researchers perform the third approach (i.e., $\text{PC}_{\text{on}\to\text{off}}$) which may be because the goal of on-policy sampling is to alleviate the out-of-distribution problem that training

| Parameter | Value | |
|---|---|---|
| | SFT | DPO |
| Epochs | 1 | 1 |
| Learning Rate | $2.0 \times 10^{-5}$ | $5.0 \times 10^{-7}$ |
| Batch size (per device) | 4 | 4 |
| Gradient Accumulation Steps | 8 | 8 |
| $\beta$ | - | 0.01 |
| warmup ratio | 0.1 | 0.1 |
| scheduler | cosine | cosine |
| GPUs | $4 \times$ A100 | $4 \times$ A100 |

Table 6: Training hyper-parameters (SFT and DPO).

| Parameter | Value |
|---|---|
| top_k | 50 |
| top_p | 0.9 |
| temperature | 0.7 |

Table 7: Inference hyper-parameters (sampling on-policy preference candidates).

on off-policy data solely suffers, but the third approach can not handle it empirically for its end up training on off-policy data. We include this setting for the completeness of the experimental setup.

### C.5 DETAILS ABOUT $\text{PC}_{\text{llama}}$

**Data Construction** To construct $\text{PC}_{\text{llama}}$, we use the raw Llama-3-8B-Instruct model to generate a pair of on-policy reference candidates, following the settings introduced in Appendix C.2 and Appendix C.4. Specifically, we use the prompts same as $\text{PC}_{\text{off}}$, which are derived from Ultra-Feedback, and annotate the preference of on-policy preference candidates by PairRM. $\text{PC}_{\text{llama}}$ and $\text{PC}_{\text{off}}$ have identical prompts but different preference candidates. We abstract the core difference between $\text{PC}_{\text{llama}}$ and $\text{PC}_{\text{off}}$ into two key characteristics, the intra-diversity and the answer quality, as introduced in §4.3. We then analysis the characteristics.

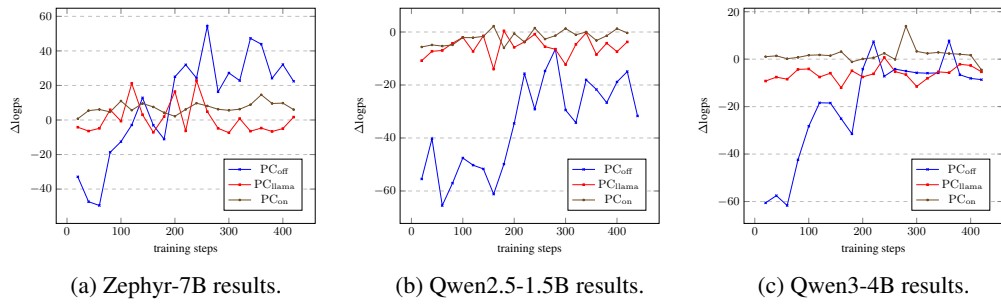

(a) Zephyr-7B results.    (b) Qwen2.5-1.5B results.    (c) Qwen3-4B results.

Figure 3: The intra-diversity between $\text{PC}_{\text{off}}$ and $\text{PC}_{\text{llama}}$ that is defined by the difference($\Delta$) of log probabilities between the chosen and the rejected answer cross different models, inclding Zephyr-7B, Qwen2.5-1.5B and Qwen3-4B. The curves of $\text{PC}_{\text{on}}$ are also included as reference.

**Diversity** This section discusses the intra-diversity between preference pairs. We define the intra-diversity as the difference between generation probability of preference pairs by a given model, operationalized by the log-probability difference between paired responses as follows:

$$Div_{\text{intra}} = \frac{1}{N} \sum_i^N (\log \pi_\theta(y_1^i|x) - \log \pi_\theta(y_2^i|x)), \tag{12}$$

where $y_1^i$ and $y_2^i$ are the chosen and the rejected answer for the $i_{th}$ sample respectively. To compare the intra-diversity between preference pairs that derived from $\mathrm{PC_{off}}$ and $\mathrm{PC_{llama}}$, we record the log probabilities of preference pairs individually when training on Zephyr-7B, Qwen2.5-1.5B, and Qwen3-4B, and present the result in Figure 3. As shown in the figure, during the training procedure, the difference in log probabilities of $\mathrm{PC_{off}}$ has a larger fluctuation range but the difference in log probabilities of $\mathrm{PC_{llama}}$ remains stable and close to zero. The results show that $\mathrm{PC_{off}}$ is more diverse than $\mathrm{PC_{llama}}$. The log probabilities of preferences pairs derived from $\mathrm{PC_{on}}$ are also included in Figure 3 as reference. As shown in the result, $\mathrm{PC_{on}}$ and $\mathrm{PC_{llama}}$ share similar trend during the training process. It indicates that though $\mathrm{PC_{llama}}$ is an off-policy preference dataset for models except for Llama-3, it still holds the low intra-diversity characteristics as an on-policy preference dataset for different models. As a result, it is a reasonable dataset for conducting comparative experiments on comparing data characteristics including intra-diversity and answer quality while avoiding their on-policy/off-policy nature.

**Quality** We define answer quality as the degree of alignment with human preference. We compare the quality by measuring the preference labeled by the ground-truth preference model between answers sampled from $\mathrm{PC_{off}}$ and $\mathrm{PC_{llama}}$. Specifically, we followed the official recipe of AlpacaEval benchmark and annotate the preference using GPT-4-turbo. The preference candidates are one randomly sampled answer from the preference candidates of $\mathrm{PC_{llama}}$ and the chosen answer of $\mathrm{PC_{off}}$, then report the result of length-controlled win rate on 805 cases that were randomly sampled from the training set. Our results show that the length-controlled (LC) win rate that answers of $\mathrm{PC_{llama}}$ being preferred is 58.84. The result shows that the quality of $\mathrm{PC_{llama}}$ is higher than that of $\mathrm{PC_{off}}$.

# D ILLUSTRATING THE RELATIONSHIP BETWEEN TEXT DISTRIBUTION ESTIMATION (§5.2) AND ALGORITHM 1 (§5.3)

We illustrate the relationship between the general text distribution estimation and our purposed boundary measurement algorithm in Figure 4.

Figure 4: Illustration of the relationship between the general text distribution estimation and our boundary measurement algorithm discussed in §5.2 and §5.3. The boundary measurement algorithm is derived from preference consistency measurement. The preference consistency measurement is purposed for estimating the consistency between two preference distributions, which are defined as proxies towards the intractable text distribution.

# E PROOFS AND DEVIATIONS

## E.1 PROOF OF THEOREM 5.1

*Proof.* Eq. (5) shows that given any reward model $r_\phi$, there is a unique policy $\pi_\theta$ that $\pi_\theta$ is the optimal solution under Eq. (3). Then, we prove that given any policy $\pi_\theta$, the corresponding reward model is unique, too.

Given $\pi_\theta$ as the optimal solution and $\pi_{\text{ref}}$ is fixed, we can transform Eq. (5) into:

$$f(x, y) = r_\phi(x, y) - \beta \log \frac{\pi_\theta(y|x)}{\pi_{\text{ref}}(y|x)} - \beta \log Z(x), \tag{13}$$

where $f(x, y)$ is always equals to zero. For some given $x_0, y_0$, we rewrite $f$ as a function of $r_\phi(x_0, y_0)$:

$$f_{x_0,y_0}(r_\phi(x_0, y_0))$$
$$= r_\phi(x_0, y_0) - \beta \frac{\pi_\theta(y_0|x_0)}{\pi_{\text{ref}}(y_0|x_0)} - \beta \log Z(x_0). \tag{14}$$

Let $r_\phi(x_0, y_0)$ be an independent variable with range $\mathcal{R}$, we can calculate the partial derivative of $f$ with respect to $r_\phi(x_0, y_0)$:

$$\begin{aligned}
&\frac{\partial f_{x_0,y_0}(r_\phi(x_0, y_0))}{\partial r_\phi(x_0, y_0)}\\
&= \frac{\partial r_\phi(x_0, y_0)}{\partial r_\phi(x_0, y_0)} - 0 - \beta \frac{1}{Z(x_0)} \frac{\partial Z(x_0)}{\partial r_\phi(x_0, y_0)}\\
&= 1 - \beta \frac{1}{Z(x_0)} \pi_{\text{ref}}(y_0|x_0) \frac{\partial \exp(\frac{1}{\beta} r_\phi(x_0, y_0))}{\partial r_\phi(x_0, y_0)}\\
&= (1 - \frac{\pi_{\text{ref}}(y_0|x_0) \exp(\frac{1}{\beta} r_\phi(x_0, y_0))}{Z(x_0)}) \frac{\partial r_\phi(x_0, y_0)}{\partial r_\phi(x_0, y_0)}\\
&= 1 - \frac{\pi_{\text{ref}}(y_0|x_0) \exp(\frac{1}{\beta} r_\phi(x_0, y_0))}{Z(x_0)}.
\end{aligned} \tag{15}$$

The partial derivative of $f$ with respect to $r_\phi(x_0, y_0)$ is always greater than or equal to zero. Due to its monotonicity, there is at most one value $r_\phi(x_0, y_0)$ that can satisfy $f(x_0, y_0) = 0$. If $\pi_{\text{ref}}$ is not a one-hot distribution (i.e., $\pi_{\text{ref}}(y_0|x_0) = 1$ and $\pi_{\text{ref}}(y|x_0) = 0$ for any $y \neq y_0$), then the range of $f$ is $\mathcal{R}$ because the domain of $r_\phi$ is $\mathcal{R}$, there will be an $r_\phi(x_0, y_0)$ that satisfies $f(x_0, y_0) = 0$. In other words, for any given $\pi_\theta$, there exists an $r_\phi$ that satisfies Eq. (5), and completes the proof of Theorem 5.1.

$\square$

## E.2 PROOF OF THEOREM 5.2

*Proof.* Let $\mathbb{P}(y_1, y_2, x) \in [0, 1]$ be the generalized form of preference that $y_1$ is preferred than $y_2$ given prompt $x$. First of all, we prove that the optimal solution of Eq. (6) satisfies for each $(x, y_1, y_2) \sim \mathcal{D}$, we have $\mathbb{P}_\phi(y_1, y_2, x) = \mathbb{P}^*(y_1, y_2, x)$. Eq. (6) can be rewritten into the following format:

$$\min_\phi \mathbb{E}_{(x,y_1,y_2)\sim\mathcal{D}}[D_{\text{kl}}(\mathbb{P}_\phi(y_1, y_2, x) \| \mathbb{P}^*(y_1, y_2, x)]. \tag{16}$$

Given that the KL divergence between two Bradley-Terry (BT) models has an exact calculation, it implies that the optimal solution for each preference pair in $\mathcal{D}$ satisfies $\mathbb{P}_\theta(y_1, y_2, x) = \mathbb{P}^*(y_1, y_2, x)$. However, we will demonstrate that $\mathbb{P}_\theta = \mathbb{P}^*$ holds only under the assumption of infinite data. Suppose that $\mathbb{P}_\theta$ is the optimal solution of Eq. (6) obtained from dataset $\mathcal{D}$. For any sample $(x, y_1, y_2) \sim \mathcal{D}$, the optimal solution ensures that $\mathbb{P}_\theta(y_1 \succ y_2|x) = \mathbb{P}^*(y_1 \succ y_2|x)$. Conversely, for any $(x, y_1, y_2) \sim \mathcal{D}'$ where $\mathcal{D}' \cap \mathcal{D} = \phi$, there is no guarantee that this equality persists, as $\mathbb{P}^*$ is unconstrained for such out-of-distribution samples. Nevertheless, under the infinite data assumption, $D$ achieves full coverage of the sample space, making $\mathcal{D}'$ an empty set. Consequently, $\mathbb{P}_\theta = \mathbb{P}^*$ holds for any $(x, y_1, y_2)$, which completes the proof of Theorem 5.2.

$\square$

### E.3  Proof of Theorem 5.4

*Proof.* We can rewrite the equation in Definition 5.3 with the following form:

$$\mathbb{P}^*(y_1 \succ y_2 | x) = \sigma(\log \frac{\pi_G(y_1|x)}{\pi_G(y_2|x)}) \tag{17}$$

Let $\mathcal{X}$ be the state space and $\mathcal{A}$ be the action space, define $f(x, y_1, y_2) : \mathcal{X} \times \mathcal{A} \times \mathcal{A} \rightarrow \mathbb{R}$ be the cocycle that for each $(x, y_1, y_2)$, the following equation holds:

$$f(x, y_1, y_2) = \frac{\pi_G(y_1|x)}{\pi_G(y_2|x)}. \tag{18}$$

Then $f$ is a fixed function given $\pi_G$. We then prove that $\pi_\theta$ which satisfies Eq. (18) does not exist unless $\pi_\theta = \pi_G$. Without loss of generality, assume there exists $\pi_\theta$ that satisfies

$$f(x, y_1, y_2) = \frac{\pi_\theta(y_1|x)}{\pi_\theta(y_2|x)}, \tag{19}$$

which is equivalence to

$$\pi_\theta(y_1|x) = f(x, y_1, y_2)\pi_\theta(y_2|x). \tag{20}$$

Let $y_2$ be a static point that has a specific value, sum $y_1$ on both sides of the equation, we have

$$\sum_{y_1} \pi_\theta(y_1|x) = \sum_{y_1} f(x, y_1, y_2)\pi_\theta(y_2|x). \tag{21}$$

Since $\pi_\theta$ is a text distribution, we have $\sum_y \pi_\theta(y|x) = 1$. Substitute the equivalence into the above equation then simplify the above formula, we have

$$\pi_\theta(y_2|x) = \frac{1}{\sum_{y_1} f(x, y_1, y_2)}. \tag{22}$$

The right hand side can be accurately calculated since the $f$ function is determined. The left hand side, which is $\pi_\theta(y_2|x)$, can be uniquely determined. And thus we prove $\pi_\theta(y_2|x) = \pi_G(y_2|x)$. Applying the result to all $y_2$, we have $\pi_\theta = \pi_G$, and completes the proof of Theorem 5.4.

$\square$

## F  Further Empirical Analysis

### F.1  Reasonableness of the Distinct Assumption

In this section, we compare the sampling probability between on-policy preference candidates and off-policy preference candidates. Since $\pi_{\text{off}}$ is intractable, we verify $\mathbb{I}[\mathbb{P}_\theta(y_1^i \succ y_2^j|x)] = 1$ and extend the result to $\mathbb{I}[\mathbb{P}_{\text{off}}(y_1^i \succ y_2^j|x)] = 0$. Specifically, we sample $2,000$ prompts from Ultra-Feedback, as well as their corresponding off-policy preference candidates and their corresponding on-policy preference candidates. For each prompt, we compare the sampling probability between one off-policy preference candidate and one on-policy preference candidate by performing a language modeling task using the corresponding policy. As for each prompt, we have two off-policy preference candidates and two on-policy preference candidates, we perform four comparisons each time, then performing a macro average and report the final win rate. The win rate is calculated as on-policy preference candidate having a higher probability than off-policy preference candidate for all the initial policy we used in our previous experiments. We provide the comparison results in Table 8. The results show that, compared to off-policy samples, initial policies assign higher probabilities to the on-policy candidates in all cases. Notably, the win rate is $84.3\% \sim 96.5\%$ for different models, indicating that our assumption is reasonable in most cases.

## G  Further Visualization Results

### G.1  System Prompt of GPT-4 Evaluation in AlpacaEval

We follow the standard recipe of the authors of AlpacaEval, where the system prompt is illustrated in Table 9.

| Iter-1 | Iter-2 | Win Rate |
|---|---|---|
| **Llama-3-8B-Instruct** | | |
| - | - | 91.06 |
| $PC_{off}$ | - | 93.97 |
| $PC_{on}$ | - | 91.11 |
| **Zephyr-7B** | | |
| - | - | 88.80 |
| $PC_{off}$ | - | 89.56 |
| $PC_{on}$ | - | 96.50 |
| **Phi-2-2.7B** | | |
| - | - | 86.96 |
| $PC_{off}$ | - | 84.32 |
| $PC_{on}$ | - | 85.89 |

Table 8: Results of the comparison between the sampling probability between $PC_{off}$ and $PC_{on}$ for different initial models. The win rate getting close to 1 shows that the initial policies assign higher probabilities to on-policy candidates.

## G.2 CASE FOR ALPACAEVAL

We provide a case from the AlpacaEval generated by Zephyr in Table 10. Though this case is neither cherry-picked nor lemon-picked, it is not randomly selected as we choose this case by its relatively short prompt length and generation length for better visualization effect.

```
<|im_start|>system
You are a highly efficient assistant, who evaluates and rank
large language models (LLMs) based on the quality of their
responses to given prompts.  This process will create a
leaderboard reflecting the most accurate and human-preferred
answers.
<|im_end|>
<|im_start|>user
I require a leaderboard for various large language models.
I'll provide you with prompts given to these models and
their corresponding responses.  Your task is to assess these
responses, ranking the models in order of preference from a
human perspective.  Once ranked, please output the results
in a structured JSON format for the make_partial_leaderboard
function.

## Prompt

{
    "instruction":  """{instruction}""",
}

## Model Outputs

Here are the unordered outputs from the models.  Each output is
associated with a specific model, identified by a unique model
identifier.

{
    {
        "model":  "m",
        "output":  """{output_1}"""
    },
    {
        "model":  "M",
        "output":  """{output_2}"""
    }
}

## Task

Evaluate and rank the models based on the quality and relevance
of their outputs.  The ranking should be such that the model
with the highest quality output is ranked first.
<|im_end|>
```

Table 9: Prompt for GPT-4 evaluation in the Alpaca Eval configuration file. Texts in blue can to be substituted by the real data.

---

**Instruction**

Complete the sentence by providing an appropriate word.

She was wearing a ___ dress.

---

**Reference Output**

The word "red" would be an appropriate word to fill in the blank in the sentence "She was wearing a [___] dress."

---

**Zephyr (SFT)**

Red

---

**Zephyr** ($PC_{off}$)

floral (adjective) dress (noun)

Confidence: 95%

---

**Zephyr** ($PC_{on}$)

Cocktail dress. She was wearing a cocktail dress.

---

**Zephyr** ($PC_{off \to off}$)

1. Flowery (if the dress had floral designs)
2. Red (if the dress was the color red)
3. Sequined (if the dress had sequins)
4. Lace (if the dress was made of lace)
5. Vintage (if the dress was from an older time period)
6. Off-shoulder (if the dress had an off-shoulder design)
7. Maxi (if the dress was very long)
8. Little black (if the dress was a simple, black, versatile dress)
9. Bohemian (if the dress had a loose, flowing style)
10. Cocktail (if the dress was appropriate for a formal event)

---

**Zephyr** ($PC_{off \to on}$)

1. Flowery: She was wearing a **floral** dress - with intricate patterns of blooming flowers.
2. Sequined: She was wearing a **sequined** dress - adorned with sparkling rhinestones and tiny beads.
(omit for 6 lines)
9. Bold-printed: She was wearing a **bold-printed** dress - featuring a striking, eye-catching pattern, such as stripes, polka dots, or abstract designs.
10. Off-shoulder: She was wearing an **off-shoulder** dress - a dress style with the neckline falling off one or both shoulders, often featuring a halter-style strap or spaghetti straps.

Hope these suggestions help! Let me know if you need more context or specificity.

---

**Zephyr** ($PC_{on \to off}$)

She was wearing a flowyairysheerdiaphanous (adjectives indicating a light and transparent fabric) maxi dress.

(Alternatively, you could also use: She was wearing a chiffonsilklaceorganza dress, depending on the specific fabric used.)

---

**Zephyr** ($PC_{on \to on}$)

Cocktail dress

Alternatively:
- Little black dress (if it was a black, form-fitting dress suitable for a formal or semi-formal occasion)
(omit for 7 lines)
- A-line dress (if the skirt flared out from the waist in a triangular shape)

---

Table 10: Responses generated by Zephyr-7b under different training iterations and trained with different preference data. We omit the outputs of **Zephyr** ($PC_{off \to on}$) and **Zephyr** ($PC_{on \to off}$).

