# OpenReview forum: "Is On-Policy Data always the Best Choice for Direct Preference Optimization-Based LM Alignment?"
_ICLR.cc/2026/Conference — ICLR 2026 Poster_

### Official Review · Reviewer_fQXn · 2025-10-31

**Soundness:** 3
**Presentation:** 2
**Contribution:** 2
**Rating:** 4
**Confidence:** 4

**Summary:**

This paper investigates whether on-policy data is always the optimal choice for DPO-based language model alignment. The authors show that the effectiveness of on-policy data varies significantly across different model architectures and scales. To explain this, they propose an alignment stage assumption that divides the alignment process into two distinct phases: the preference injection stage, where diverse data is more beneficial, and the preference fine-tuning stage, where high-quality data is preferred. The authors further propose an algorithm to identify the boundary between these two stages.

**Strengths:**

- The paper provides the important observation that on-policy data can sometimes degrade alignment performance.

- The authors empirically demonstrate that the effectiveness of on-policy versus off-policy data varies across model scales and architectures, offering nuanced insights into LM alignment dynamics.

- The notion of preference injection highlights the mid-training importance of data diversity.

**Weaknesses:**

- Some theoretical sections (especially §5.2–§5.3) are mathematically dense and could benefit from more intuitive explanations, visualizations, or pseudocode examples.

- The boundary measurement algorithm is under-emphasized. It would be better if it were moved from the appendix to the main text.

- The paper could include more comparative experiments across different model families in the **main body** to validate the proposed assumption and algorithm further.

**Questions:**

- In Table 2, is the diversity of PC_on higher than that of PC_llama?

- Could you briefly summarize the purpose and contribution of Section 5.1?

- Could the boundary measurement algorithm be used to adaptively schedule on-policy and off-policy sampling in large-scale industrial alignment pipelines? I found that the computational complexity of the proposed algorithm appears relatively high.

- The paper mainly evaluates on AlpacaEval 2.0 (length-controlled) — have you tested whether similar trends hold on MT-Bench, Arena-Hard, or other evaluation suites?

- In practice, high-quality data often has low diversity. Have you explored data blending strategies to balance these two properties?

---

> ### Author Response · Authors · 2025-11-22
> **(1/4) Response to Reviewer fQXn**
>
> Dear Reviewer fQXn,
>
> We sincerely thank you for the constructive comments and suggestions, which are very helpful for improving our paper. We are also grateful that you recognized the strengths of our paper. Please kindly find point-to-point responses below.
>
> > **W1:(Clarity of Theoretical Section)** Some theoretical sections (especially §5.2–§5.3) are mathematically dense and could benefit from more intuitive explanations, visualizations, or pseudocode examples.
>
> Thanks for the suggestion. We have added an illustration about §5.2 and §5.3 in **Figure 4, Appendix D** in our revised manuscript.
>
> > **W2:(Position of Boundary Measurement Algorithm)** The boundary measurement algorithm is under-emphasized. It would be better if it were moved from the appendix to the main text.
>
> Thanks for the suggestion. We have moved the boundary measurement algorithm into the main text **(Section 6)** in our revised manuscript.
>
> > **W3:(Comparative Experiments in the Main Body)** The paper could include more comparative experiments across different model families in the main body to validate the proposed assumption and algorithm further.
>
> Thanks for the suggestion. For all results of comparative experiments, we have moved them into the main body **(Section 6)**  in our revised manuscript to improve the integrity.
>
> > **Q1:(Diversity between PC_on and PC_llama)** In Table 2, is the diversity of PC_on higher than that of PC_llama?
>
> Thanks for your question. We evaluate the diversity of PC_on and that of PC_llama by different models including Zephyr, Qwen2.5-1.5B and Qwen3-4B, and the visualization result can be found in Appendix C.5.
>
> The results show that for different models, the log-probability difference between paired response of PC_on is more stable and closer to zero than that of PC_llama, which indicates that the diversity of PC_on is lower than that of PC_llama. However, the difference on intra-diversity between PC_on and PC_llama is relatively less significant compared with that between PC_on and PC_off, or PC_llama and PC_off, in which case we prefer treating the diversity of PC_on and PC_llama as the similiar level.

---

> ### Author Response · Authors · 2025-11-22
> **(2/4) Response to Reviewer fQXn**
>
> > **Q2:(Purpose of Section 5.1)** Could you briefly summarize the purpose and contribution of Section 5.1?
>
> Thanks for your question. Briefly, in Section 5.1, we introduce theorem 5.1 and theorem 5.2, aiming at clarifying the equivalence between the DPO objective (Eq.6) and the RL objective (Eq.3), and providing the motivation of our further discussion about text distribution in Section 5.2.
>
> Specifically, We talk about the bijection between reward model and policy in theorem 5.1, which is a theoretical supplement to DPO to make sure the equivalence between the DPO objective (Eq.6) and the RL objective (Eq.3). In [1], the authors established a one-way mapping from reward function r* to an optimal policy $\pi*$. However, DPO optimizes a causal policy $\pi_\theta$ during its pactical optimization process, instead of the optimial policy $\pi*$, and it is risky to assume that a causal policy $\pi_\theta$ satisfies the one-way mapping between $\pi*$ and r* (i.e., Eq.5). For example, while optimizing the policy via Eq.6 (Eq.7 in [1]), in the intermediate step, the being-optimized policy $\pi_\theta$ may not satisfy Eq.5 (Eq.4 in [1]). Then, although it is a practically acceptable approach towards LM alignment, the DPO objective is not equivalent with the RL objective. In theorem 5.1, we show that for any policy $\pi_\theta$, there exists a unique reward model that satisfies Eq.3. In that case, for each intermediate step, the being-optimized policy $\pi_\theta$ satisfies Eq.5, and the DPO objective and RL objective are equivalent, solidifying its theoretical foundation.
>
> Theorem 5.2 establishes a necessary condition for achieving the optimal solution of the RL objective (Eq.3). This directly bridges the theoretical optimum of DPO with the practical concern of preference data selection. Based on this theorem, we proceed in Sections 5.2 and 5.3 to estimate the underlying text distribution, which is a distribution defined by the reward model that reflects the characteristics of an ideal, infinite dataset.
>
> [1] Rafailov et al. "Direct Preference Optimization: Your Language Model is Secretly a Reward Model." NeurIPS2023
>
> > **Q3:(Computational Cost of Boundary Measurement Algorithm)** Could the boundary measurement algorithm be used to adaptively schedule on-policy and off-policy sampling in large-scale industrial alignment pipelines? I found that the computational complexity of the proposed algorithm appears relatively high.
>
> We thank the reviewer for raising this important and practical question regarding the computational complexity of our boundary measurement algorithm and its applicability in industrial-scale pipelines. We agree that any method intended for large-scale deployment must be computationally efficient. As detailed in Appendix B.1, the computational overhead of our boundary measurement is estimated to be **3.2%** of a single training epoch.
>
> We also perform a quantitative experiment to compare the time required for the boundary measurement algorithm and the alignment process. Specifically, a complete alignment process can be divided into the following stages:
>
> * Preference Sampling (on-policy only). To obtain on-policy data, the current policy is required to generate responses given a set of prompts.
>
> * Preference Labeling. Given a pair of preference candidates, a reward/preference model is asked to label preference between the preference candidates.
>
> * DPO Optimization. The policy is trained via DPO method on the labeled preference dataset.
>
> We perform the experiment for training Zephyr-7B in PC_on on 4 A100-40G GPUs. The time required for each stage is:
>
> | Procedures | Time |
> | :- | :-: |
> | Preference Sampling | 4.5hrs |
> | Preference Labeling | 3.5hrs |
> | DPO Optimization | 8.9hrs|
> | Total | **16.9hrs** |
>
> and the time required for boundary measurement algorithm is:
>
> | Procedures | Time |
> | :- | :-: |
> | Preference Sampling | 9mins |
> | Boundary Measurement | 14mins |
> | Total | **23mins** |
>
> The results show that the boundary measurement algorithm can schedule on-policy and off-policy sampling in a cheap and proactive way.

---

> ### Author Response · Authors · 2025-11-22
> **(3/4) Response to Reviewer fQXn**
>
> > **Q4:(More Evaluation Suites)** The paper mainly evaluates on AlpacaEval 2.0 (length-controlled) — have you tested whether similar trends hold on MT-Bench, Arena-Hard, or other evaluation suites?
>
> Thank you for raising this important point regarding the generalizability of our findings across different evaluation benchmarks. We agree that validating our results on multiple suites is crucial for demonstrating the robustness of the alignment stage assumption. We perform additional experiments for the three main models: Zephyr, Llama-3 and Phi-2 on MT-Bench to evaluate our stage assumption. The results are as follows:
>
> The results of Llama-3:
>
> | Iter1  | Iter2  | Turn 1 | Turn 2 | Avg. |
> | :-    | :-    | :-: | :-: | :-: |
> | -      | -      | 7.0375 | 6.4750 | 6.7563 |
> | PC_off | -      | 7.7375 | 6.2250 | 6.9813 |
> | PC_on  | -      | 7.9000 | 6.5875 | **7.2438** |
> | PC_off | PC_off | 7.4250 | 6.3625 | 6.8938 |
> | PC_off | PC_on  | 7.6000 | 6.4625 | **7.0300** |
> | PC_on  | PC_off | 7.4875 | 6.4000 | 6.9438 |
> | PC_on  | PC_on  | 7.7125 | 6.5625 | **7.1375** |
>
> The results of Zephyr-7B:
>
> | Iter1 | Iter2 | Turn 1 | Turn 2 | Avg. |
> | :- | :- | :-: | :-: | :-: |
> | -      | -      | 5.3375 | 4.2625 | 4.8000 |
> | PC_off | -      | 6.2250 | 5.5750 | **5.9000** |
> | PC_on  | -      | 5.9875 | 5.0000 | 5.4938 |
> | PC_off | PC_off | 6.2250 | 5.4625 | 5.8438 |
> | PC_off | PC_on  | 6.3375 | 5.5875 | **5.9625** |
> | PC_on  | PC_off | 6.1625 | 5.5000 | **5.8313** |
> | PC_on  | PC_on  | 5.8000 | 5.1875 | 5.4938 |
>
>
> The results of Phi-2:
>
> | Iter1 | Iter2 | Turn 1 | Turn 2 | Avg. |
> | :- | :- | :-: | :-: | :-: |
> | - | - | 5.0625 | 3.6750 | 4.3688 |
> | PC_off | - | 5.6000 | 3.9375 | **4.7688** |
> | PC_on  | - | 5.2625 | 3.9125 | 4.5875 |
> | PC_off | PC_off | 6.0000 | 4.5625 | **5.2813** |
> | PC_off | PC_on  | 5.7250 | 4.4125 | 5.0688 |
> | PC_on  | PC_off | 5.7375 | 4.3875 | **5.0625** |
> | PC_on  | PC_on  | 5.3375 | 3.8750 | 4.6063 |
>
> As shown in the results:
>
> * For **llama-3**, for all initial checkpoints, model trained on PC_on achieves a better performance on MT-Bench compared with the model trained on PC_off.
>
> * For **Zephyr**, for initial checkpoints including sft / trained on PC_on in the first iteration, model trained on PC_off acheieves a better performance on MT-Bench compared with model trained on PC_on. For initial checkpoint where the model is trained on PC_off in the first iteration, model trained on PC_on achieves a better performance on MT-Bench compared with model trained on PC_off.
>
> * For **Phi-2**, for all initial checkpoints, model trained on PC_off achieves a better performance on MT-Bench compared with the model trained on PC_on.
>
> The results on MT-Bench are observed to have similar trends compared with the results on Alpaca Eval, showing that our conclusion is robust and well-supported.

---

> ### Author Response · Authors · 2025-11-22
> **(4/4) Response to Reviewer fQXn**
>
> > **Q5:(Data Blending Strategies)** In practice, high-quality data often has low diversity. Have you explored data blending strategies to balance these two properties?
>
> We thank the reviewer for this forward-looking question. We fully agree that adaptive blending is a promising direction. While our current paper focuses on establishing the alignment stage assumption and the boundary measurement algorithm, we conduct a preliminary experiment with a simple blending strategy, PC_mix, which mixes on-policy and off-policy candidates.
>
> Specifically, the preference candidates in PC_mix are sampled from both on-policy preference candidates and off-policy preference candidates. Given the preference candidates from PC_on and PC_off with regard to the same prompt, we sample one response from PC_on and one from PC_off, then label the preference by the preference model PairRM.
>
> The results on Zephyr-7B are as follows:
>
> | Dataset | LC Win Rate | Win Rate |
> | :- | :-: | :-: |
> | PC_on   | 33.28       | 36.85    |
> | PC_off  | 23.77       | 21.67    |
> | PC_mix  | 25.47       | 24.14    |
>
> As shown in the results, model trained in PC_mix performs better than that trained in PC_off, and performs worse than PC_on.
>
> * **Observation 1:** On-policy high-quality data can be more effective for models in the preference fine-tuning stage, which strengthens our conclusion.
>
> * **Observation 2:** The data blending strategy can greatly influence the alignment effect and model performance, and the potential of designing dynamic data blending strategies during the model optimization process.
>
> As we state in the Conclusion and Limitation section (Section 7), our stage assumption inspires exploration of smoother and more adaptive data blending strategies rather than a rigid switch. Thanks again for your suggestion, we see our work as providing the tool that makes advanced, adaptive blending strategies possible and principled.

---

### Official Review · Reviewer_j12w · 2025-10-31

**Soundness:** 3
**Presentation:** 3
**Contribution:** 3
**Rating:** 6
**Confidence:** 4

**Summary:**

This paper investigates when and why on-policy preference data (generated during training) improves or harms language model alignment compared to static preference data. While previous work such as DPO and SLiC-HF assumes that on-policy sampling always benefits alignment, the authors empirically find that this is not universally true, where performance varies systematically across models (e.g., Llama-3, Zephyr) and alignment stages. To explain these findings, they introduce the alignment stage assumption, distinguishing between a preference injection stage that benefits from data diversity and a preference fine-tuning stage that benefits from data quality. They provide both theoretical analysis and empirical evidence supporting this two-stage perspective, along with an algorithm to identify the boundary between stages. Experiments across multiple model families and alignment methods demonstrate the generality and practical usefulness of this framework.

**Strengths:**

1.  The writing is clear and easy to follow.

2. Research on the problem of using on-policy data for DPO is interesting.

3. The authors have great literature review for related works.

**Weaknesses:**

1. I have some concerns about models to measure diversity. As mentioned in the paper, the authors use Zephyr to measure diversity. To further confirm this conclusion, it's better for authors to see whether this diversity patterns are similar for different models.

2. This method can be used to recognize the training stage of LLMs. Does this method have great potential to make DPO training more efficient (like convergence speed)?

3. I find that the authors conduct some experiments on DPO and SLiC-HF. However, some of iterative DPO algorithms which use on-policy data have different loss objectives as DPO or SLiC-HF. Does this method still work on that iterative DPO algorithm like [1]?

[1] Self-play preference optimization for language model alignment.

**Questions:**

Please refer to weaknesses part.

---

> ### Author Response · Authors · 2025-11-22
> **(1/2) Response to Reviewer j12w**
>
> Dear Reviewer j12w,
>
> Thank you for your thoughtful and constructive feedback and comments! We deeply appreciate your suggestions and spare no effort during this response stage to make improvements accordingly. We hope our responses below could address your concerns:
>
> > **W1:(Diversity Measurement)** I have some concerns about models to measure diversity. As mentioned in the paper, the authors use Zephyr to measure diversity. To further confirm this conclusion, it's better for authors to see whether this diversity patterns are similar for different models.
>
> Thank you for the helpful suggestion. We agree that testing across different model families and sizes can better demonstrate the diversity measurement. To this end, we conducted experiments on two additional models: Qwen2.5-1.5B and Qwen3-4B, and the visualization results are shown in **Figure 3** in **Appendix D.5**. As shown in the results, the diversity of off-policy dataset is consistently higher than PC_llama across different models.
>
> We appreciate this suggestion, as it has helped us better assess the robustness of our experiments. We will include these results in the final version.
>
>
> > **W2:(Potential of More efficient Training)** This method can be used to recognize the training stage of LLMs. Does this method have great potential to make DPO training more efficient (like convergence speed)?
>
> We thank the reviewer for this insightful question. Actually, one purpose of the alignment stage assumption and boundary measurement algorithm is to make DPO training more efficient, particularly in terms of convergence speed and sample efficiency.
>
> Our method improves efficiency primarily by dynamically selecting the most effective data at each model state, thereby avoiding wasted computation on sub-optimal data. Taking the result of Zephyr in Table 3 as an example. Given the similar computational resources, training zephyr for two iteration by "PC_on fist, PC_off second", or by "PC_off first, PC_on second" can result in significant differences with regard to model performances.
>
> Specifically, the LC win rate for zephyr trained by "PC_on first, PC_off second" is 22.22, while that for zephyr trained by "PC_off first, PC_on second" is 33.28. This substantial performance gap underscores the importance of using the right data at the right time, which can be modeled by the alignment stage assumption, and estimated by the boundary measurement algorithm. The characteristics of alignment stages, i.e., quality and diversity, can also be helpful for constructing suitable preference datasets for different models.

---

> ### Author Response · Authors · 2025-11-22
> **(2/2) Response to Reviewer j12w**
>
> > **W3:(Generalizability towards Iterative DPO Variants)** I find that the authors conduct some experiments on DPO and SLiC-HF. However, some of iterative DPO algorithms which use on-policy data have different loss objectives as DPO or SLiC-HF. Does this method still work on that iterative DPO algorithm like SPPO?
>
> Thank you for raising this important point regarding the generalizability of our proposed alignment stage assumption and boundary measurement algorithm to iterative DPO variants with different loss objectives, such as SPPO. We conduct experiments on SPPO with three different models: Zephyr, Phi-2 and LLama-3. All settings are aligned with the main experiments in our manuscript.
>
> | Iter1 | Iter2 | LC Win Rate | Win Rate | BS (initial) |
> | :- | :- | :-: | :-: | :-: |
> | - | - | 5.81 | 3.72 | - | - |
> | PC_off | - | **5.89** | 3.92 | 0.23 |
> | PC_on  | - | 5.25 | 3.21 | 0.23 |
> | PC_off | PC_off | **8.08** | 7.11 | 0.38 |
> | PC_off | PC_on  | 6.88 | 6.23 | 0.38 |
> | PC_on  | PC_off | **7.15** | 4.85 | 0.36 |
> | PC_on  | PC_on  | 6.92 | 4.13 | 0.36 |
>
> | Iter1 | Iter2 | LC Win Rate | Win Rate | BS (initial) |
> | :- | :- | :-: | :-: | :-: |
> | - | - | 8.12 | 4.25 | - |
> | PC_off | - | **13.16** | 11.29 | 0.40 |
> | PC_on  | - | 10.93 | 7.16 | 0.40 |
> | PC_off | PC_off | 14.94 | 12.28 | 0.56 |
> | PC_off | PC_on  | **20.12** | 18.23 | 0.56 |
> | PC_on  | PC_off | **17.33** | 13.40 | 0.43 |
> | PC_on  | PC_on  | 13.42 | 9.38  | 0.43 |
>
> | Iter1 | Iter2 | LC Win Rate | Win Rate | BS (initial) |
> | :- | :- | :-: | :-: | :-: |
> | - | - | 24.59 | 24.47 | - | - |
> | PC_off | - | 26.83 | 25.16 | 0.62 |
> | PC_on  | - | **33.94** | 35.38 | 0.62 |
> | PC_off | PC_off | 27.54 | 26.31 | 0.65 |
> | PC_off | PC_on  | **36.17** | 40.74 | 0.65 |
> | PC_on  | PC_off | 34.81 | 36.05 | 0.69 |
> | PC_on  | PC_on  | **38.64** | 45.23 | 0.69 |
>
> As shown in the results, Llama-3 is in preference fine-tuning stage in all experiments, Zephyr is in preference fine-tuning stage after being trained on PC_off in the first iteration, while Phi-2 is in preference injection stage in all experiments. **All experimental results are aligned with the results of boundary measurement algorithm**, which shows that our method is robust and well-supported.

---

### Official Review · Reviewer_NSmT · 2025-11-01

**Soundness:** 3
**Presentation:** 3
**Contribution:** 3
**Rating:** 6
**Confidence:** 4

**Summary:**

The authors finds that On-policy data for alignment is not always optimal and attempts to explain it with dividing the alignment stage into a "preference injection" stage which requires data coverage (diversity) and a preference fine-tuning stages (which requires high quality data). The authors verify their findings on different models and alignment algorithms (DPO, SLiC-HF).

**Strengths:**

1. The paper investigates the learning dynamics of preference learning, which is understudied. The findings are novel and could be useful to practitioners who are working on human preference learning.

2. The paper is well written with comprehensive experiments demonstrating the generalizability of their findings. The finding that off-policy DPO followed by on-policy DPO on two iterations provides a simple recipe for people to try out.

3. The discussion between on- v.s. off-policy data could be extended to many other settings that involves Language Model post-training.

**Weaknesses:**

1. It seems that people are not really excited about DPO / RLHF anymore. For example, the latest open-source frontier models (Qwen3, GLM 4.5, Kimi-K2...) only adopts a RL process using rubric rewards and verifiable rewards on math / coding tasks. This is a good and interesting paper, it is just that I don't know how much impact would a DPO paper make in 2025. It would be better if the authors can show  a curriculum of on- and off-policy data also generalizes to RLVR experiments.

2. The findings are somewhat validated on, in my understanding, weak models. (e.g. Zephyr-7B and Phi-2-2.7B). I don't know if the base model is stronger (e.g. Qwen3 series) how much the findings would still hold. It seems to me that the on-policy data is not good simply because that some of the reference model were not good enough to produce reasonable outputs, so we need to reply on off-policy good demonstrations to perform alignment.

**Questions:**

See above.

---

> ### Author Response · Authors · 2025-11-22
> **(1/2) Response to Reviewer NSmT**
>
> Dear Reviewer NSmT,
>
> Thanks for your comprehensive and detailed suggestions for our work! We really value your comment on generalization of our findings towards RLVR scenario and the scale of our conclusions. We hope our response could address your concerns:
>
>
> >**W1:(Generalization towards RLVR)** It seems that people are not really excited about DPO / RLHF anymore. For example, the latest open-source frontier models (Qwen3, GLM 4.5, Kimi-K2...) only adopts a RL process using rubric rewards and verifiable rewards on math / coding tasks. This is a good and interesting paper, it is just that I don't know how much impact would a DPO paper make in 2025. It would be better if the authors can show a curriculum of on- and off-policy data also generalizes to RLVR experiments.
>
> We thank the reviewer for your insightful comment and for recognizing the value of our work. We fully agree that the trend towards RLVR in the latest frontier models is a significant and powerful direction for enhancing performance in mathematically verifiable domains like coding and math.
>
> Regarding the suggestion to extend our experiments to the RLVR setting, we have given it careful consideration. We believe that while the core intuition of our work is broadly relevant, a direct application to a typical RLVR setup presents fundamental challenges due to a paradigm mismatch. Please allow us to humbly elaborate on our perspective.
>
> 1. The value/application of preference-based alignment in 2025.
>
> Preference-based alignment focuses on scenarios where **the judgment standard is based on complex, nuanced human values and intentions that are difficult to define with simple rules or a single correct answer**. To improve model capabilities like helpfulness and harmlessness, preference-based methods are still the recommended recipe (e.g. Preference Alignment in 4.4 General RL in [1]).
>
> 2. The difference between preference-based alignment and RLVR.
>
> Our work is intrinsically centered on preference-based alignment methods like DPO, which rely on preference signal where rewards follow the Bradley-Terry definition and the preference is derived from the comparison between a pair of response sequences. In contrast, for rubric and verifiable rewards that are used in RLVR tasks, the model's response / action quality is evaluated in a **standardized judge system**, where the rewards are sparse and discrete (e.g., 0/1 indicating whether the final answer is correct or incorrect). The reward does not follow the Bradley-Terry definition. Hence, it is difficult to define the preference between two answers. For instance, it is challenging to define a meaningful preference between two incorrect answers, which is essential for our framework but not supported by a typical verifier's binary signal.
>
> 3. Valueable Preference is one of the main challenges in RLVR.
>
> Our work is based on preference model, which can provide clear and meaningful preference between the preference candidates, and is a fundmantal concept in preference alignment tasks. However, for RLVR tasks, clearly distinguishing the relative advantage given a group of trajectories is an ongoing research question, especially for optimization methods like GRPO. Actually, several works have already been trying to incorporate dynamic sampling and entropy control to ensure the generated trajectories can be assigned different rewards[2][3]. The difficulty of preference definition and preference representation is a key limitation that prevents this work from extending to RLVR scenarios.
>
>
> However, we fully agree with the reviewer's comment that the discussion of data effiency is universal. For example, research on offline reinforcement learning focuses on utilizing previously collected data to optimize models, especially when on-policy trajectories are difficult, expensive or dangerous to acquire. **We have added the discussion about curriculum learning strategies for on- and off-policy data in scenarios beyond preference-based alignment in Appendix B.6 in the revised manuscript.**
>
>
> [1] Yang, et al. "Qwen3 Technical Report." arXiv preprint arXiv:2505.09388 (2025).
>
> [2] Yu, et al. "DAPO: An Open-Source LLM Reinforcement Learning System at Scale." arXiv preprint arXiv:2503.14476 (2025).
>
> [3] Zheng, et al. "Group Sequence Policy Optimization." arXiv preprint arXiv:2507.18071 (2025).

---

> ### Author Response · Authors · 2025-11-22
> **(2/2) Response to Reviewer NSmT**
>
> >**W2: (Clarity and Scale of our Findings)** The findings are somewhat validated on, in my understanding, weak models. (e.g. Zephyr-7B and Phi-2-2.7B). I don't know if the base model is stronger (e.g. Qwen3 series) how much the findings would still hold. It seems to me that the on-policy data is not good simply because that some of the reference model were not good enough to produce reasonable outputs, so we need to rely on off-policy good demonstrations to perform alignment.
>
> Thank you for this insightful and constructive comment. The question of how our findings generalize to more powerful foundation models is indeed critical, and we appreciate the opportunity to clarify our core contribution and address this point directly.
>
> > It seems to me that the on-policy data is not good simply because that some of the reference model were not good enough to produce reasonable outputs, so we need to rely on off-policy good demonstrations to perform alignment.
>
> We fully agree that model capability is a key part of the phenomenon we observed, and the observation is elegantly explained by our proposed Alignment Stage Assumption. The weaker initial capability of models like Zephyr and Phi-2 can be a primary reason for the inferior performance of their on-policy data. Intuitively, aligning a model with "good" data should be more efficient than using data that is not "good" enough. If the on-policy is "good", then we should use on-policy data; otherwise we should use off-policy data. However, **it is difficult to define what "good" data is, and this intuition is highly empirical and relatively vague**. This leads to the core question we want to answer in this paper, i.e., the requirements for preference candidates during the LM alignment process.
>
> Generally, we move beyond this observation to provide a generalizable framework that explains why data effectiveness shifts and provides actionable tools to predict it. Specifically, we propose the Alignment Stage Assumption, focus on intra-diversity and answer quality as the key characteristics of the alignment stages, and aim to identify the "good" data during the LM alignment process.
>
> We kindly refer the results of experiments in Section 4.3, where some interesting observations may be helpful to address your concern. As shown in Table 2, when we train Zephyr on PC_llama, a high-quality off-policy dataset for Zephyr (which can also provide off-policy good demonstrations), the performance is still worse than that trained on PC_off, a relatively low-quality but high-diversity off-policy dataset for Zephyr. The result shows that performing alignment on high-quality datasets may be sub-optimal for models in preference injection stage, in which case the high-diversity datasets can be more useful. What's more, the PC_llama dataset can be more efficient when Zephyr enters the preference fine-tuning stage. These experimental results indicate that the data requirement and the definition of "good" data (model preference for diversity / quality) are dynamic during the LM alignment process.
>
> >The findings are somewhat validated on, in my understanding, weak models. (e.g. Zephyr-7B and Phi-2-2.7B). I don't know if the base model is stronger (e.g. Qwen3 series) how much the findings would still hold.
>
> As mentioned above, our conclusion is not that *off-policy data is always better*, but *the more efficient preference candidates are dynamic and depend on the model's current alignment stage*. Following our framework, we would predict that a powerful base model (like the Qwen3 series), especially if it has undergone high-quality instruction tuning, would likely begin the DPO alignment process firmly in the preference fine-tuning stage, just as Llama-3 in the main experiments. This is further verified by our experiments.
>
> Specifically, we perform the two-iteration experiments on Qwen3-4B. The results are shown below.
>
> | Iter1 | Iter2 | LC Win Rate | Win Rate | BS (initial) |
> | :- | :- | :-: | :-: | :-: |
> | - | - | 33.51 | 40.62 | - |
> | PC_off | - | 29.49 | 32.54 | 0.67 |
> | PC_on | - | **37.53** | 43.44 | 0.67 |
> | PC_off | PC_off | 23.77 | 22.34 | 0.67 |
> | PC_off | PC_on | **37.21** | 42.83 | 0.67 |
> | PC_on | PC_off | 31.42 | 33.03 | 0.71 |
> | PC_on | PC_on | **39.36** | 44.74 | 0.71 |
>
> As shown in the results, for all initial checkpoints, the model trained on PC_on performs better than that trained on PC_off, showing that Qwen3-4B is in the preference fine-tuning stage, a trend similar to that observed with Llama-3 in the main paper. The boundary scores are consistent across all conditions, indicating that our boundary measurement algorithm remains valid for more powerful models. Overall, **the results with Qwen3 align with the conclusions presented in the main paper**.

---

### Official Review · Reviewer_cNBy · 2025-11-03

**Soundness:** 2
**Presentation:** 3
**Contribution:** 3
**Rating:** 6
**Confidence:** 4

**Summary:**

The submission proposes an qualitative breakdown of the preference optimization into the preference injection and preference fine-tuning stage, based on varied effectiveness of on- and off-policy preference data usage on different models. Authors validate the proposed breakdown on multiple models and datasets, and propose a quantitative boundary area measurement to determine the preference stage a given model and preference dataset pair is in.

**Strengths:**

- Taking inititative on the varied results we see in the preference optimization literature is good to see. The breakdown of the preference optimization at least makes sense qualitatively, if not theoretically
- The boundary area measurement provides a single quantitative metric that really distills the core of the paper
- The research questions are validated on multiple model families

**Weaknesses:**

- While the boundary area measurement is a good starting point, I still think the proposition given in the paper lacks practical application. The boundary area measurement is easy to calculate in retrospect, after we have the models and all versions of the preference data, but doesn't seem so easy when decisions need to be made on-the-fly.

**Questions:**

- What are some practical applications of the propositions made in the paper, when it comes to deciding between on- and off-policy data for preference optimization?

---

> ### Author Response · Authors · 2025-11-22
> **Response to Reviewer cNBy**
>
> Dear Reviewer cNBy,
>
> Thanks for your comprehensive and detailed suggestions for our work! We really value your comment on the applicability and practical application of our conclusions. We hope our response could address your concerns:
>
> >**W1:(Applicability)** While the boundary area measurement is a good starting point, I still think the proposition given in the paper lacks practical application. The boundary area measurement is easy to calculate in retrospect, after we have the models and all versions of the preference data, but doesn't seem so easy when decisions need to be made on-the-fly.
>
> We sincerely thank the reviewer for this insightful point. We agree that for any methodological contribution to have a significant impact, it must be actionable within a practical training pipeline.
>
> We agree with your comment that our *experimental setup* in the paper was retrospective for the purpose of a controlled ablation (the full-combination two-iteration experiment). However, **the algorithm itself is not retrospective, but lightweight and forward-passing**.
>
> To determine the alignment stage of a given model, one should perform preference sampling on a small set, then perform the boundary measurement algorithm we proposed. If the boundary score indicates that the model is in preference injection stage, then the off-policy preference dataset should be more effective than the on-policy preference dataset. One can decide the suitable preference dataset **prospectively**, instead of creating the on-policy preference dataset, performing DPO training and running evaluation.
>
> Actually, our discussion on computational cost of boundary measurement algorithm in Appendix B.1 shows that the data required for performing boundary measurement algorithm is estimated to be **3.2%** of a single training epoch. To present the differences in computational cost more intuitively, we perform the experiment for training Zephyr-7B in PC_on on 4 A100-40G GPUs. The time required for each stage is:
>
> | Procedures | Time |
> | :- | :-: |
> | Preference Sampling | 4.5hrs |
> | Preference Labeling | 3.5hrs |
> | DPO Optimization | 8.9hrs|
> | Total | **16.9hrs** |
>
> and the time required for boundary measurement algorithm is:
>
> | Procedures | Time |
> | :- | :-: |
> | Preference Sampling | 9mins |
> | Boundary Measurement | 14mins |
> | Total | **23mins** |
>
> The results show that the boundary measurement algorithm can schedule on-policy and off-policy sampling in a cheap and proactive way.
>
> Furthermore, our alignment stage assumption and analysis about the characteristics of alignment stages have potential practical value, which can help researchers and practitioners understand the data requirements of specific models and tailor their data strategy accordingly, making the alignment process more efficient.
>
> >**Q1:(Practical Application)** What are some practical applications of the propositions made in the paper, when it comes to deciding between on- and off-policy data for preference optimization?
>
> We thank the reviewer for this important question. Our work provides a general framework for dynamically selecting the suitable preference data during preference optimization. Our work can help researchers and practitioners understand the dynamic data requirements for LMs during the alignment process, and thus improve the alignment effectiveness from the perspective of preference data improvement. We list some practical applications as follows:
>
> * Efficient and Cost-Effective Training. As on-policy sampling is computationally expensive, our method reduces the need for frequent on-policy data collection. For models in the preference injection stage, practitioners can rely more on off-policy data throughout training, saving significant computational resources.
>
> * Principled Preference Data Construction. As we analyzed quality and diversity as the two key characteristics of the alignment stage, researchers and practitioners can purposefully construct preference datasets. For instance, during the preference injection stage, one should prioritize highly diverse data; during the preference fine-tuning stage, the focus should shift to high-quality, high-reward preference data. This provides actionable dimensions for data construction.

---

### Author Response · Authors · 2025-11-25
**General Response**

We sincerely thank all the reviewers for their thoughtful comments and constructive suggestions, which significantly helped us strengthen our paper. We are encouraged to see that the reviewers recognize the significance of our research question (Reviewer cNBy, NSmT, j12w, fQXn), the novelty of proposed Alignment Stage Assumption and Boundary Measurement Algorithm (Reviewer cNBy, j12w, fQXn), its sound design and comprehensive experimental validation (Reviewer cNBy, NSmT, fQXn). We present the general concerns and the summary of the paper revision below. We hope this summary helps the AC's efforts in making the final decision.

### **General Concerns**

The reviewers' general concerns primarily focused on the following points:

* **Applicability & Practical Application:** Reviewers concerned about the computational cost of the boundary measurement algorithm, and whether the algorithm can be applied on-the-fly. We show that the computational overhead of our algorithm is estimated to be 3.2% of a single training epoch, and can be applied prospectively.

* **Generalizability of Conclusions:** Reviewers asked about the generalizability of our conclusions. We perform experiments on additional LM (Qwen3-4B), additional alignment algorithm (SPPO) and additional evaluation suite (MT-bench) to show our conclusions are robust and well-supported.

* **More LMs towards Diversity Measurement:** Reviwers suggested performing more LMs to measure the diversity between preference candidates. We perform experiments on Qwen2.5-1.5B and Qwen3-4B to show the diversity patterns are similar for different models.

In response to the reviewers' feedback, we have submitted an updated version of our paper, which now includes more experimental details, more visualizations about our theoretical section, and additional discussion about related works. Below, we summarize the revisions made to the paper.

### **Summary of Paper Revision**

* **[Section 5 & Appendix D]** We update the introduction of the theoretical section, and add visualizations to help better understanding of our theoretical sections (especially section 5.2 & 5.3).

* **[Section 5]** We move the Boundary Measurement Algorithm from appendix to the main paper to emphasize its importance and improve the integrity.

* **[Section 6]** We move the generalizability analysis about additional LMs and additional alignment algorithm from appendix to the main paper to improve the integrity of our paper.

* **[Appendix C.5]** We add more visualization results about diversity measurement across different models including Zephyr, qwen2.5 an qwen3 to better assess the robustness of our experiments.

* **[Appendix D]** We add additional discussion about on- and off-policy curriculum learning strategies about more general domains including offline reinforcement learning and RLVR tasks.

We would like to express our sincere gratitude to all reviewers for their valuable tirle and effort in helping improve our work, and we would like to check whether you have any additional questions or concerns that we can help clarify.

Best regards,

Authors

---

### Meta-Review · Area_Chair_8P8V · 2026-01-03

**Summary:**

This paper investigates a timely and important question for preference-based language model alignment: Is on-policy data always the best choice for Direct Preference Optimization (DPO)? The authors present both empirical and theoretical evidence showing that on-policy data can sometimes be detrimental, and propose the Alignment Stage Assumption, which decomposes alignment into a preference injection stage emphasizing diversity and a preference refinement stage emphasizing quality. They further introduce a lightweight boundary measurement algorithm to identify the transition between stages and guide data selection.

Reviewers generally agreed that the research question is novel and well motivated, and that the findings challenge a commonly held assumption in the alignment literature. While there were initial concerns regarding practical impact, scope, and positioning relative to recent alignment trends, the rebuttal substantially strengthened the paper by clarifying motivation, expanding experiments, and better situating the contribution. After discussion, the overall sentiment shifted toward viewing the work as a valuable analytical and empirical contribution that improves understanding of DPO behavior, even if it does not propose a new alignment algorithm.

Given the originality of the question, the clarity of the empirical evidence, and the improved presentation after rebuttal, the paper meets the bar for ICLR acceptance.

**Reviewer Concerns:**

Concerns addressed by the rebuttal:

•	Generality across models and methods:
Additional experiments on Qwen models, SPPO, and MT-Bench convincingly demonstrate that the observed phenomenon is not limited to a single model or DPO variant.

•	Practical overhead of the boundary measurement algorithm:
The rebuttal showed that the algorithm incurs minimal computational cost and can be applied prospectively, alleviating concerns about usability.

•	Clarity and exposition:
The authors reorganized the paper, added clearer explanations and figures, and improved the accessibility of both the theoretical and empirical sections.

Remaining but non-blocking concerns:

•	Relevance to broader alignment pipelines:
While some reviewers noted that frontier models increasingly rely on RLVR-style approaches, the community still actively studies DPO and preference optimization; the paper’s insights are therefore relevant and complementary.

•	Actionability:
The work is more diagnostic and analytical than algorithmic, but this is appropriate given the paper’s stated goal of understanding data effects rather than proposing a new optimizer.

Overall, remaining concerns are primarily about scope rather than correctness or significance, and do not preclude acceptance.

**Reviewer Scores:**

•	Reviewer cNBy: Likely improves from 6 → 7, positive and supportive, with no blocking concerns.

•	Reviewer NSmT: Likely improves from 6 → 7, generally favorable after rebuttal clarifications.

•	Reviewer j12w: Remains at 6, views the contribution as insightful and well supported.

•	Reviewer fQXn: Likely improves from 4 → 5, with improved clarity and additional experiments addressing earlier concerns.

---

### Decision · Program_Chairs · 2026-01-26

Accept (Poster)